# Pharmacological polyamine catabolism upregulation with methionine salvage pathway inhibition as an effective prostate cancer therapy

Hayley C. Affronti[1,9], Aryn M. Rowsam[1,9], Anthony J. Pellerite[1], Spencer R. Rosario[1], Mark D. Long [1], Justine J. Jacobi[1], Anna Bianchi-Smiraglia [2], Christoph S. Boerlin[1], Bryan M. Gillard[3], Ellen Karasik[3], Barbara A. Foster[3], Michael Moser [3], John H. Wilton[3], Kristopher Attwood[4], Mikhail A. Nikiforov [2], Gissou Azabdaftari[5], Roberto Pili[6], James G. Phillips[7], Robert A. Casero Jr.[8] & Dominic J. Smiraglia [1]*

Prostatic luminal epithelial cells secrete high levels of acetylated polyamines into the prostatic lumen, sensitizing them to perturbations of connected metabolic pathways. Enhanced flux is driven by spermidine/spermine N1-acetyltransferase (SSAT) activity, which acetylates polyamines leading to their secretion and drives biosynthetic demand. The methionine salvage pathway recycles one-carbon units lost to polyamine biosynthesis to the methionine cycle to overcome stress. Prostate cancer (CaP) relies on methylthioadenosine phosphorylase (MTAP), the rate-limiting enzyme, to relieve strain. Here, we show that inhibition of MTAP alongside SSAT upregulation is synergistic in androgen sensitive and castration recurrent CaP models in vitro and in vivo. The combination treatment increases apoptosis in radical prostatectomy ex vivo explant samples. This unique high metabolic flux through polyamine biosynthesis and connected one carbon metabolism in CaP creates a metabolic dependency. Enhancing this flux while simultaneously targeting this dependency in prostate cancer results in an effective therapeutic approach potentially translatable to the clinic.

[1] Department of Cancer Genetics and Genomics, Roswell Park Comprehensive Cancer Center, Buffalo, NY 14263, USA. [2] Department of Cell Stress Biology, Roswell Park Comprehensive Cancer Center, Buffalo, NY 14263, USA. [3] Department of Pharmacology and Therapeutics, Roswell Park Comprehensive Cancer Center, Buffalo, NY 14263, USA. [4] Department of Biostatistics, Roswell Park Comprehensive Cancer Center, Buffalo, NY 14263, USA. [5] Department of Pathology, Roswell Park Comprehensive Cancer Center, Buffalo, NY 14263, USA. [6] Department of Hematology and Oncology, Indiana University, Indianapolis, IN, USA. [7] Department of Translational Hematology and Oncology Research, Taussig Cancer Institute, Cleveland Clinic, Cleveland, OH 44195, USA. [8] The Sydney Kimmel Comprehensive Cancer Center, Johns Hopkins University, Baltimore, MD 21231, USA. [9] These authors contributed equally: Hayley C. Affronti, Aryn M. Rowsam. *email: Dominic.Smiraglia@roswellpark.org

Despite improvements to therapies targeting the androgen axis, prostate cancer (CaP) remains the leading cause of cancer related incidence and the third cause of cancer related mortality in men in the United States[1]. Patients that present with recurrent CaP are treated with androgen deprivation therapy (ADT)[2]. However, inevitably patients develop castration-resistant prostate cancer (CRPC)[2]. Thus, there remains a need for therapies to treat CaP more effectively. Most current strategies aim to better target the androgen axis at multiple points. Abiraterone acetate, enzalutamide[2], and, more recently, apalutamide[3] have demonstrated benefit, but alternative and more effective therapies independent of the androgen axis remain necessary[4].

Cancer therapies that target specific genetic alterations face many challenges due to the high degree of genomic heterogeneity that exists among patients and intratumorally[5]. Genetic and epigenetic alterations can be heterogeneous, but those changes often have phenotypic convergence on the same or highly related metabolic pathways[6]. Therefore, targeting a cancer metabolic phenotype that is downstream of multiple common genetic changes offers a potentially fruitful alternative therapeutic direction. Therapies that target metabolism were initially generated as single agents and drug resistance commonly developed[7,8]. Combination strategies that target metabolic vulnerabilities from multiple angles may provide better therapies less likely to result in acquisition of resistance. Understanding unique, tissue-specific metabolic dependencies may provide opportunities to identify therapeutic windows using strategies that aim to leverage those dependencies.

The high rate of secretion of acetylated polyamines into the prostatic epithelial lumen makes prostate epithelial cells exquisitely sensitive to perturbations of connected pathways[9–13]. This enhanced polyamine metabolic flux is driven by the activity of spermidine/spermine N1-acetyltransferase (SSAT) which acetylates polyamines to facilitate their secretion into the lumen[14–16]

(Fig. 1). SSAT activity drives polyamine catabolism resulting in intracellular polyamine pool deficits that are replenished by upregulation of polyamine biosynthesis, or by back conversion through polyamine oxidase and spermine oxidase (PAOX and SMOX)[17]. In addition to the role secreted polyamines play in prostatic fluid, intracellular polyamines are important for numerous cellular functions[8,17–19]. Importantly, CaP cells maintain secretion of polyamines, but also are proliferative and therefore require high levels of intracellular polyamines. This leads to increased demand on connected metabolic pathways to support this flux. Previous studies have demonstrated that all polyamine levels were higher in CaP than benign prostatic hyperplasia[20]. Therefore, a metabolic vulnerability may exist as a result of polyamine biosynthetic flux, and polyamine metabolism or other connected metabolic pathways may be targeted to control CaP. Furthermore, the lower levels of flux through polyamine biosynthesis and catabolism in non-prostate tissues may limit toxicity, which is a significant advantage created by leveraging a tissue-specific vulnerability.

The high level of polyamine biosynthesis in prostate places added strain on connected pathways, including one-carbon metabolism composed of the methionine cycle and folate metabolism, which are critical for production of SAM and nucleotide pools, respectively[9–13]. Prostate cells are sensitive to dietary folate deficiency in the transgenic adenoma of mouse prostate cancer (TRAMP) model[11] and the CWR22 human xenograft model of CRPC[13], due to increased acetylated polyamine export. The methionine salvage pathway (MSP) plays a critical role in helping to relieve metabolic strain by recycling the one-carbon unit, 5′-methylthioadenosine (MTA), lost to polyamine biosynthesis back to the methionine cycle (Fig. 1). The rate-limiting enzyme involved in this process is methylthioadenosine phosphorylase (MTAP), and human CaP cell lines and xenografts are sensitive to MTAP inhibition or knockdown[9].

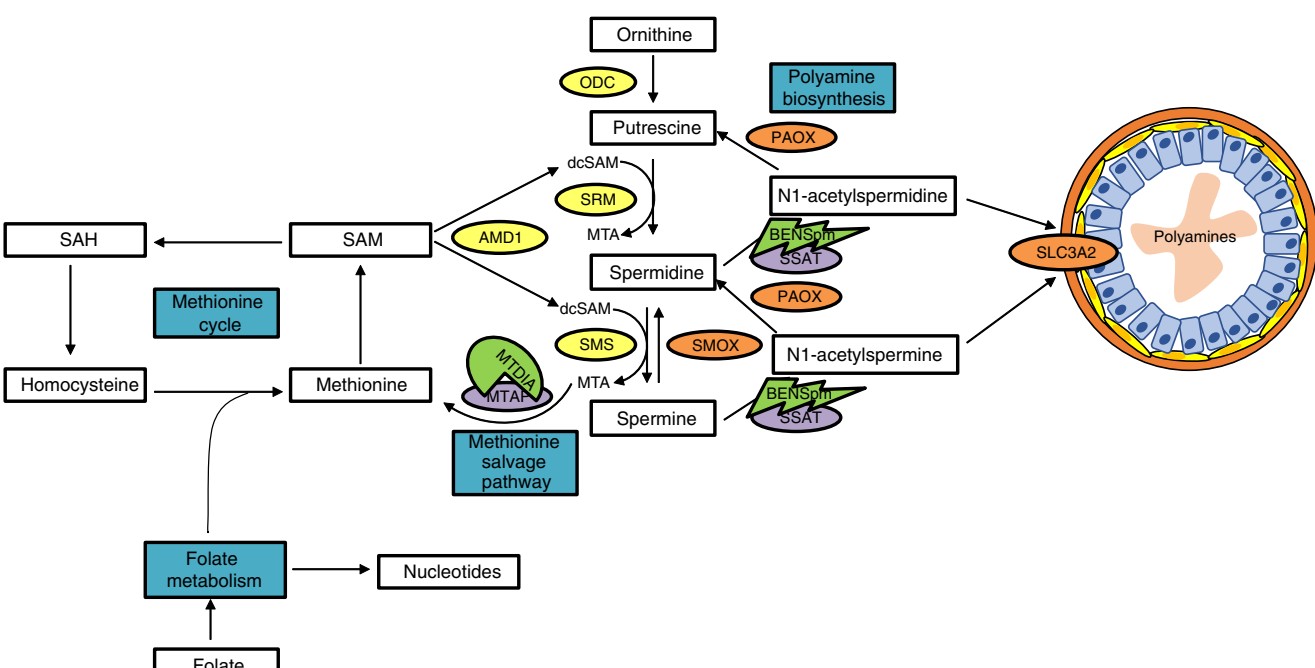

**Fig. 1 Overview of BENSpm and MTDIA Therapeutic Approach.** Pathways are highlighted in blue. Target Enzymes are highlighted in purple; methylthioadenosine phosphorylase (MTAP) and spermidine/spermine N[1]-acetyltransferase (SSAT). Pharmacological agents are highlighted in green and next to their corresponding target; Methylthio-DADMe-Immucillin-A (MTDIA) and N[1],N[11]-bisethylnorspermine (BENSpm). Poylamine biosynthesis enzymes are highlighted in yellow; ornithine decarboxylase (ODC1), s-adenosylmethionine decarboxylase (AMD1), spermidine synthase (SRM) and spermine synthase (SMS). The acetylated polyamine exporter is highlighted in pink; solute carrier 3 family member 2 (SLC3A2).

The current study tests the hypothesis that enhancing metabolic stress by driving acetylated polyamine secretion above the already high levels found in CaP, while simultaneously blocking the ability of the cell to mitigate this stress by inhibiting the MSP, will impact connected pathways and cause cell death. Methylthio-DADMe-Immucillin-A (MTDIA), a transition state analog inhibitor of MTAP, is combined with the well characterized polyamine analog, $N^1$-$N^{11}$-bisethynorspermine (BENSpm), that increases SSAT activity enhancing acetylated polyamine export (Fig. 1). Herein, we report that MTDIA and BENSpm cause a synergistic block in proliferation and increase in cell death in a panel of CaP cell lines that include both the androgen-sensitive and androgen-independent states. Additionally, MTDIA and BENSpm treatment impact intracellular polyamine levels, SAM and SAH pools, and reactive oxygen species production. Finally, these treatments are effective in CaP xenograft models and ex vivo patient samples. This combination therapy leverages the extraordinary levels of flux through polyamine metabolism in CaP cells and may provide an improved therapeutic approach.

## Results

**BENSpm and MTDIA synergize in prostate cancer cell lines.** $N^1,N^{11}$-bisethylnorspermine (BENSpm) and Methylthio-DADMe-Immucillin-A (MTDIA) have been used as single agents in a variety of cancer cell lines, with IC50s ranging from 10 to 100 nM for MTDIA[21,22] and 1.0 μM to 1 mM for BENSpm[23,24]. BENSpm was administered safely in phase I/II clinical trials at a dose of 100 mg/M² [25,26]. Since this combination has not previously been reported we first evaluated whether there was a synergistic or additive relationship between the two drugs. A range of doses from 1 nM to 10 μM MTDIA and 100 nM to 2.5 μM BENSpm were used to test for synergism in three CaP cell lines in the presence of 20 μM MTA.

Combination therapy was tested in the androgen-sensitive cell lines LNCaP and LAPC-4, and two androgen-independent lines, LNCaP-C4-2 (C4-2) and CWR22Rv1. LNCaP has a mutated AR that allows for response to DHT[27] and additional ligands while LAPC-4 has a wild-type AR[28]. Both are highly responsive to androgen stimulation. C4-2 and CWR22Rv1 have mutated AR. BENSpm significantly decreased the effective dose of MTDIA in LNCaP, C4-2, and CWR22Rv1 (Fig. 2a–c) cells. CWR22Rv1 cells were the most sensitive to addition of BENSpm, where the IC50 was lowered by greater than 10,000-fold. The Chou-Talalay[29] method produced Combination Indexes <1.0 that indicated a synergistic relationship between MTDIA and BENSpm (Fig. 2d). FACS sorting of Annexin-V and Propidium Iodide stained LNCaP, C4-2, CWR22Rv1, and LAPC-4 cells revealed a decreased number of live cells and an increased number of early and late apoptotic cells with treatment (Fig. 2e). These experiments proved that BENSpm and MTDIA treatment was synergistic at blocking cell proliferation, and induced cytotoxicity following 8 days of treatment, additively.

To determine the effect of treatments on the target enzymes MTAP and SSAT, LNCaP and C4-2 cells were treated with MTDIA and BENSpm alone or in combination. Cells were treated for 8 days in the presence of 20 μM MTA with 1 nM MTDIA and 1 μM BENSpm (a synergistic dose with respect to growth inhibition in LNCaP, C4-2, and CWR22Rv1) for the remainder of the mechanistic studies. BENSpm or the combination treatment resulted in an approximately 7x increase in SSAT activity (pmol/min/mg) in LNCaP cells (Fig. 2f – black bars) which confirmed previous findings[30]. The fact that SSAT gene expression is enhanced by androgen stimulation raised the question of whether or not BENSpm would be effective in a castrate environment. Importantly, an approximately 10x increase in SSAT activity (Fig. 2g – black bars) was found in androgen-independent C4-2 cells in charcoal-stripped serum. BENSpm treatment resulted in SSAT activities of 949 and 756 pmol/min/mg in LNCaP and C4-2, respectively, demonstrating that it is effective in both the androgen replete environment and in the absence of androgens.

Knocking down SSAT should rescue the anti-proliferative effect and eliminate synergy with MTDIA since the effect of BENSpm is mediated by increasing SSAT activity. SSAT was stably knocked down using two short hairpin RNAs (shRNAs – shSSAT A and shSSAT B). Both shRNAs partially knocked down SSAT at the mRNA and protein levels (Supplementary Fig. 1B, C) and resulted in decreased SSAT activity relative to scramble control cells treated with BENSpm or the combination (Fig. 2f, g – gray bars vs back bars). Notably, some inducible SSAT activity persisted after BENSpm treatment in the SSAT knockdown lines, which indicated incomplete knockdown. Nevertheless, SSAT knockdown significantly rescued growth in combination treated LNCaP and C4-2 cells, (Fig. 2h, i) although not completely, which is consistent with the incomplete suppression of SSAT activity (Fig. 2f, g).

**Therapy reduces polyamines in androgen-sensitive CaP cells.** Both MTDIA and BENSpm treatment might be expected to reduce intracellular polyamine pools. MTDIA leads to a build-up of MTA, which can inhibit polyamine synthases, while BENSpm induces SSAT activity that drives polyamine catabolism and export of acetylated polyamines. Intracellular polyamines, BENSpm, and secreted acetylated polyamines were measured using Ultra Performance Liquid Chromatography (UPLC) following an 8-day treatment with vehicle control, 1 nM MTDIA, 1 μM BENSpm, or the combination. BENSpm levels were the highest in LNCaP cells while LAPC-4, CWR22Rv1, and C4-2 cells all had approximately 3-4x less BENSpm accumulation (Supplementary Fig. 2A). BENSpm enters the cell through polyamine import and previous findings have revealed that LNCaP cells, unlike other cell lines, maintain polyamine import following treatment with BENSpm[30], which may explain higher levels of BENSpm in LNCaP. Treatment with BENSpm or the combination significantly decreased intracellular spermidine and spermine levels (Fig. 3a) and BENSpm treatment increased the spermidine-to-spermine ratio in LNCaP (Supplementary Fig. 2B). The spermidine-to-spermine ratio was also increased with MTDIA treatment alone in LNCaP (Supplementary Fig. 2B). In contrast, intracellular polyamine levels and ratios in C4-2 and CWR22Rv1 were unaffected by treatments (Fig. 3a and Supplementary Fig. 2C). In accordance with previous findings[30], BENSpm and/or the combination treatment increased extracellular acetylated polyamines in all 4 cell lines, although this response was enhanced in the androgen-sensitive cell lines (Fig. 3b). In agreement with these findings, the RNA expression of the acetylated polyamine exporter, SLC3A2, was also significantly increased following BENSpm and combination treatment in LNCaP and LAPC-4 cells, but not in C4-2 or CWR22Rv1 (Fig. 3c). ODC and AMD1 activity were not significantly affected by treatments in both LNCaP and C4-2 cells (Supplementary Fig. 2D–G). Consistent with the polyamine findings in LNCaP and LAPC-4, the SAM-to-SAH ratio was decreased in the androgen-sensitive but not the androgen-independent cell lines (Fig. 3d), which may reflect increased consumption of SAM for polyamine biosynthesis. Despite similar synergy between MTDIA and BENSpm in the androgen-sensitive and androgen-independent lines (Fig. 2a–d) there was a difference in how intracellular polyamines and SAM pools were affected by the treatments. Although an increase in acetylated polyamine export

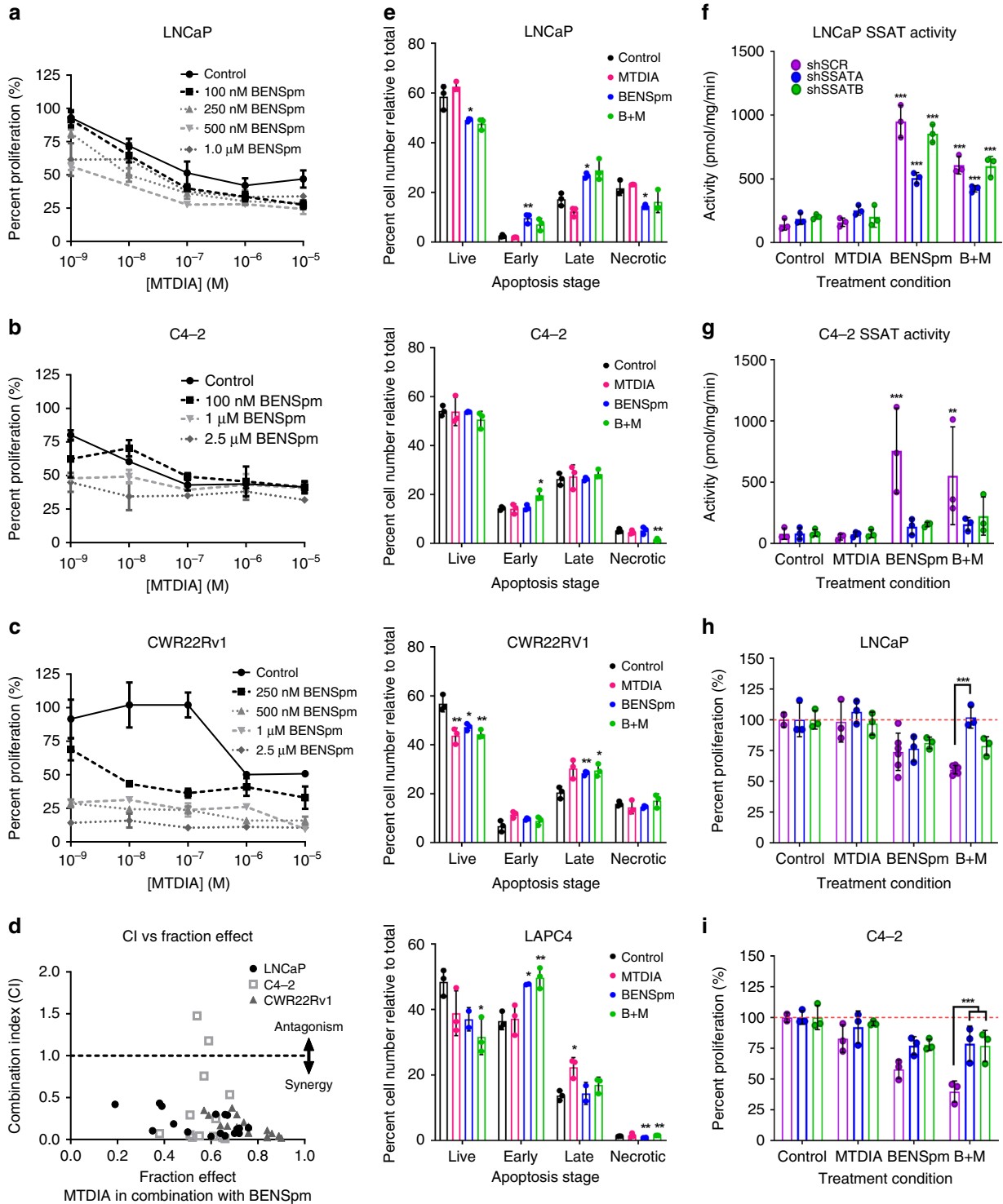

was seen in all cell lines, export may be higher in androgen-sensitive cell lines, leading to reductions in intracellular polyamine and SAM pools to a greater extent than the androgen-independent lines.

**Polyamine catabolism upregulation increases ROS production.**
BENSpm induces spermine oxidase (SMOX) resulting in increased production of hydrogen peroxide ($H_2O_2$) and the aldehyde, 3-aminopropanal in addition to its effect on SSAT activity and exportation of acetylated polyamines. The acetylated

polyamines produced by SSAT can be substrates for polyamine oxidase (PAOX) that also produces $H_2O_2$ and the aldehyde 3-aceto-aminopropanal. It is important to note that this increase in reactive oxygen species (ROS) and toxic aldehydes is concurrent with the depletion of spermidine and spermine, which can act as free radical scavengers[31,32]. The increase in ROS can play an important role in BENSpm's anti-proliferative effects in breast cancer models[23].

BENSpm and the combination treatment increased SMOX and PAOX activity in C4-2, LNCaP, and LAPC-4 (Fig. 4a, b). Furthermore, PAOX activity was increased following MTDIA

**Fig. 2 BENSpm and MTDIA treatment is synergistic.** Cell proliferation curves for **a** 1 androgen sensitive cell line, LNCaP and 2 androgen independent lines, **b** C4-2 and **c** CWR22Rv1 treated with a vehicle control, 1 nM to 10 μM MTDIA, 100 nM to 2.5 μM BENSpm, or the combination of MTDIA and BENSpm at the indicated doses all in the presence of 20 μM MTA. Number of live cells were counted by trypan blue exclusion and reported relative to vehicle control, to give percent proliferation. For the remainder of the studies, cells were treated with vehicle control, 1 nM MTDIA, 1 μM BENSpm, or the combination (B + M) in 20 μM MTA for 8 days. **d** Combination Index values vs. the fraction effected for all doses shown in graphs 1**a**–**c**. Combination Index (CI) values calculated using the Chou-Talaly method and CompuSyn for LNCaP, C4-2, and CWR22Rv1 cells treated with BENSpm and MTDIA at indicated doses (CI < 1.0 = synergistic, CI > 1.0 = antagonistic, CI = 1.0 is additive). **e** Two androgen sensitive cell lines, LNCaP and LAPC-4 and two androgen independent cell lines, C4-2 and CWR22Rv1, were analyzed after 8 days of combination treatment (1 μM BENSpm and 1 nM MTDIA) by Flow Cytometry following staining with Annexin-V and Propidium iodide. The number of live cells, early apoptotic, late apoptotic, and necrotic cells are shown for all cell lines. SSAT enzyme activity assayed for LNCaP (**f**) and C4-2 (**g**) cells transfected with a scramble control (shSCR – black bars), or one of two shRNAs to *SSAT* (shSSAT A- blue bars or B- green bars). Indicated specific enzymatic activity is reported as pmol of radiolabeled acetyl-CoA produced per minute relative to protein concentration (pmol/minute/mg of protein). Percent cell proliferation (relative to vehicle control for each shRNA) for LNCaP (**h**) and C4-2 (**i**). Results for biological triplicates are shown (*n* = 3). Statistical analyses were performed using an unpaired Student *t*-test with Welch's correction. All values are compared to vehicle control. Error bars for **a–c** represent standard error of the mean and for E-I represent standard deviation of the mean. *$p < 0.05$, **$p < 0.01$, ***$p < 0.001$.

treatment in LNCaP cells (Fig. 4b). Interestingly, the fold change was higher for enzymes in C4-2 cells (Fig. 4a, b). Similar increases were also seen in LAPC-4 treated cells, where, like LNCaP, MTDIA increased catabolic enzyme activity (Fig. 4a, b). For CWR22Rv1, BENSpm treatment also resulted in increased PAOX and SMOX activity, whereas the increase was not significant in the combination treated cells which suggested MTDIA may counteract these effects in CWR22Rv1 cells at this time point. ROS (specifically $H_2O_2$ as measured by Amplex Red) was increased in BENSpm and combination treated LNCaP, C4-2, and CWR22Rv1 cells (Fig. 4c – blue bars). MTDIA treatment alone also resulted in increased ROS production in LNCaP cells. These data indicated that ROS is upregulated following treatment in all cell lines, and suggested that this may be due to increased PAOX and SMOX activity. Furthermore, when we knocked down *SMOX* and *PAOX* in LNCaP cells using two targeting shRNAs each, we find that knockdown of *SMOX* resulted in reduced basal levels and treatment induced ROS after 8 days of treatment with either vehicle control, 1 nM MTDIA, 1 μM BENSpm, or the combination. Although we were able to knockdown PAOX at the protein level, PAOX enzyme activity was maintained, consistent with the comparable ROS levels between non-silencing control and knockdown of *PAOX* cells (Supplementary Fig. 3). This suggests that in LNCaP, ROS induction with BENSpm and combination treatment is at least in part due to enhanced SMOX activity. Interestingly, the CWR22Rv1 cell line, which is the most sensitive to treatments (Fig. 2c) had both higher baseline levels of ROS and the highest levels after treatment. A mitochondrial ROS clean-up enzyme, thioredoxin reductase 2 (*TXNRD2*), was overexpressed to determine if ROS played a role in the treatments' anti-proliferative effect[33]. LNCaP, C4-2, and CWR22Rv1 cells were transduced with lentiviral vectors which either expressed *TXNRD2*, or an empty control (PLVP) (Supplementary Fig. 4A). *TXNRD2* overexpression reduced ROS accumulation completely in C4-2, and partially in CWR22Rv1, (Fig. 4c – blue vs purple bars). However, *TXNRD2* overexpression did not reduce ROS accumulation in LNCaP cells. *TXNRD2* overexpression rescued growth completely in C4-2 but did not rescue growth in LNCaP cells which confirmed earlier findings (Fig. 4d – blue bars vs purple bars). A partial rescue was seen in CWR22Rv1 cells treated with BENSpm (Fig. 4d), with no significant differences seen in the combination. These findings indicated that increased ROS levels contributed to the anti-proliferative response to BENSpm and combination treatment in C4-2 and CWR22Rv1 cells, but were inconclusive concerning LNCaP.

The inability of *TXNRD2* overexpression to reduce ROS accumulation in LNCaP is striking. TXNRD2 functions to reduce oxidized thioredoxins and proteins, thereby acting as a ROS scavenger in the mitochondria to maintain the REDOX homeostasis[33]. Prior studies have shown that C4-2 cells have a higher antioxidant capacity and extracellular and intracellular Redox state than LNCaP[34]. Furthermore, the fold change of ROS accumulation was higher in LNCaP than C4-2 cells following BENSpm and the combination treatment (Fig. 4c). Therefore, the lower antioxidant capacity and higher accumulation of ROS in LNCaP cells, might explain why *TXNRD2* overexpression was unable to rescue the ROS production and the anti-proliferative effect. LNCaP and C4-2 PLVP and *TXNRD2* overexpressing cells were treated with 25, 50, 75, and 100 μM $H_2O_2$ for 24 h in the presence of 20 μM MTA and ROS accumulation along with proliferation rate were measured. *TXNRD2* overexpression significantly diminished ROS accumulation and rescued cell proliferation at 25 and 50 μM $H_2O_2$ in LNCaP cells, however at 75 and 100 μM $H_2O_2$ *TXNRD2* overexpression failed to block ROS accumulation and failed to rescue of proliferation (Supplementary Fig. 4B, C). In contrast, *TXNRD2* overexpression reduced ROS accumulation and rescued cell proliferation at all $H_2O_2$ concentrations tested in C4-2 cells (Supplementary Fig. 4D, E). Interestingly, treatment with BENSpm and combination induced ROS levels that exceed that of 100 μM $H_2O_2$ treatment in LNCaP cells. Together this suggested that the threshold at which *TXNRD2* overexpression can rescue ROS accumulation is higher in C4-2 cells, and that the failure to rescue in LNCaP is due to their lower antioxidant capacity and the high level of ROS induced.

**BENSpm and MTDIA block CWR22Rv1 xenograft growth**. A three month toxicity study was performed in immunocompetent Balb/C mice. MTDIA was given continuously in the drinking water throughout the study and BENSpm was given at the indicated doses using intraperitoneal (i.p.) injection (Supplementary Fig. 5A). BENSpm was given once daily, for 5 days on followed by two-weeks off. No toxicities were seen by change in body weight (Supplementary Fig. 5A), organ weights (Supplementary Fig. 5B–E), blood chemistry, or CBC.

The efficacy of the drug combination or single agents was tested in the castration-resistant setting using CWR22Rv1 androgen-independent xenografts. Xenografts were grown subcutaneously in castrated male Athymic/Nude mice with no addition of testosterone. 1 million CWR22Rv1 cells were injected into the right flanks and allowed to reach 300 to 400 mm³. Notably, the range of starting tumor volumes was ~3–6 times larger than other therapeutic studies using this model[35,36]. Animals were randomly assigned to 1 of 4 treatment groups; vehicle control, 50 mg/kg MTDIA, 100 mg/kg BENSpm

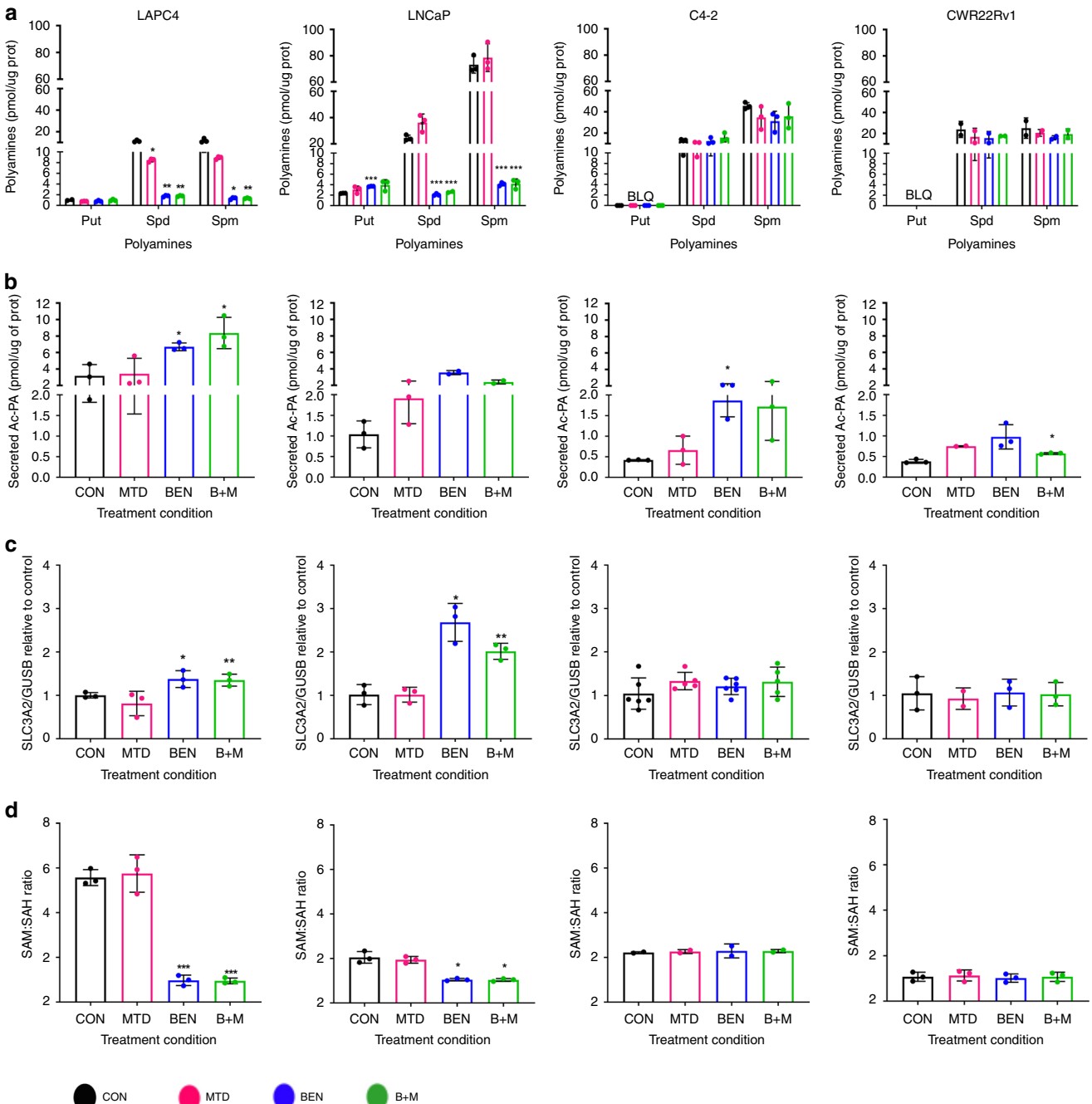

**Fig. 3 Impacts on Polyamine Biosynthesis in androgen sensitive and androgen independent lines.** LAPC-4, LNCaP, C4-2, and CWR22Rv1 cells were treated with vehicle control (black bars), 1 nM MTDIA (red bars), 1 μM BENSpm (blue bars), or the combination (B + M) (green bars) for 8 days in the presence of 20 μM MTA. **a** Intracellular putrescine, spermidine, and spermine levels measured by UPLC for all 4 cell lines. Values are normalized to protein concentrations. **b** Extracellular acetylated polyamines (acetylated spermidine + acetylated spermine) extracted from the media measured by UPLC. Values are normalized to intracellular protein concentrations. **c** RNA expression as measured by Real-Time rt-PCR for solute carrier 3 family member 2 (*SLC3A2*). Values are normalized to *GUSB* expression and made relative to vehicle control. **d** Intracellular s-adenosylmethionine (SAM) to s-adenosylhomocysteine (SAH) ratio as measured by UPLC. Results for biological triplicates are shown. Statistical analyses were performed using an unpaired student t-test with Welch's correction. All values are compared to vehicle control. Error bars represent the standard deviation of the mean. $*p < 0.05$, $**p < 0.01$, $***p < 0.001$.

or 50 mg/kg MTDIA and 100 mg/kg BENSpm (B + M). MTDIA was available continuously in the drinking water, while BENSpm was given twice weekly by i.p. injection. Animals were sacrificed once tumors grew to >2 cm³ or following 6 weeks of treatment. No toxicities were observed. BENSpm and the combination significantly reduced tumor growth rates (Fig. 5a), prolonged days to progression (Fig. 5b), and decreased tumor weights at sacrifice

(Fig. 5c). Most notably the combination resulted in the longest time to reach 400% the original tumor volume (Fig. 5b). Strikingly, the combination treatment was more effective than either drug alone as evident by the tumor weights at sacrifice (Fig. 5c). We applied the Bliss Combination Index[37] approach for both tumor weight and growth inhibition rates to ask if the combination therapy was additive or synergistic. Based on this

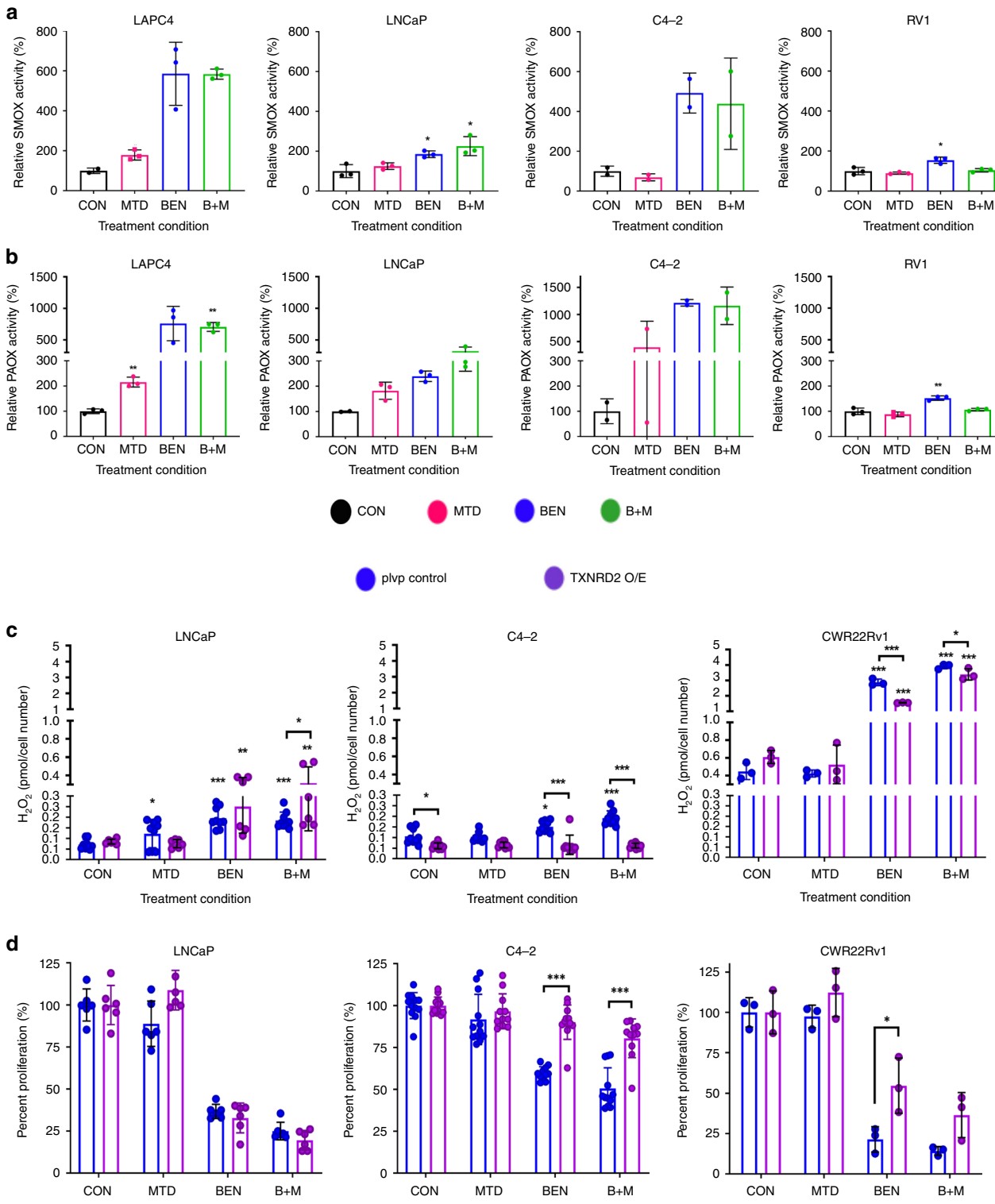

**Fig. 4 BENSpm and the combination enhance polyamine catabolism and reactive oxygen species production.** Cells were treated with vehicle control (black bars), 1 nM MTDIA (red bars), 1 μM BENSpm (blue bars) or the combination (B + M) (green bars) for 8 days in the presence of 20 μM MTA. **a** SMOX and **b** PAOX enzyme activity. Values are indicative of the amount of $H_2O_2$ produced normalized to protein concentration, relative to vehicle controls. Statistical analyses were performed using an unpaired Student's *t*-test with Welch's correction. **c** The amount of reactive oxygen species (ROS) produced normalized to cell number in cells containing vector control (PLVP) (blue bars) or thioredoxin reductase 2 (*TXNRD2*) overexpression (O/E) (purple bars) for LNCaP, C4-2, and CWR22Rv1 cells. **d** Percent cell proliferation (relative to vehicle control for each condition - PLVP or *TXNRD2* O/E) for LNCaP, C4-2, and CWR22Rv1. Results for biological triplicates are shown. Statistical analyses were performed using an unpaired Student's *t*-test with Welch's correction. All values are compared to vehicle control unless otherwise indicated by connecting lines. Error bars represent standard error of the mean. *$p < 0.05$, **$p < 0.01$, ***$p < 0.001$.

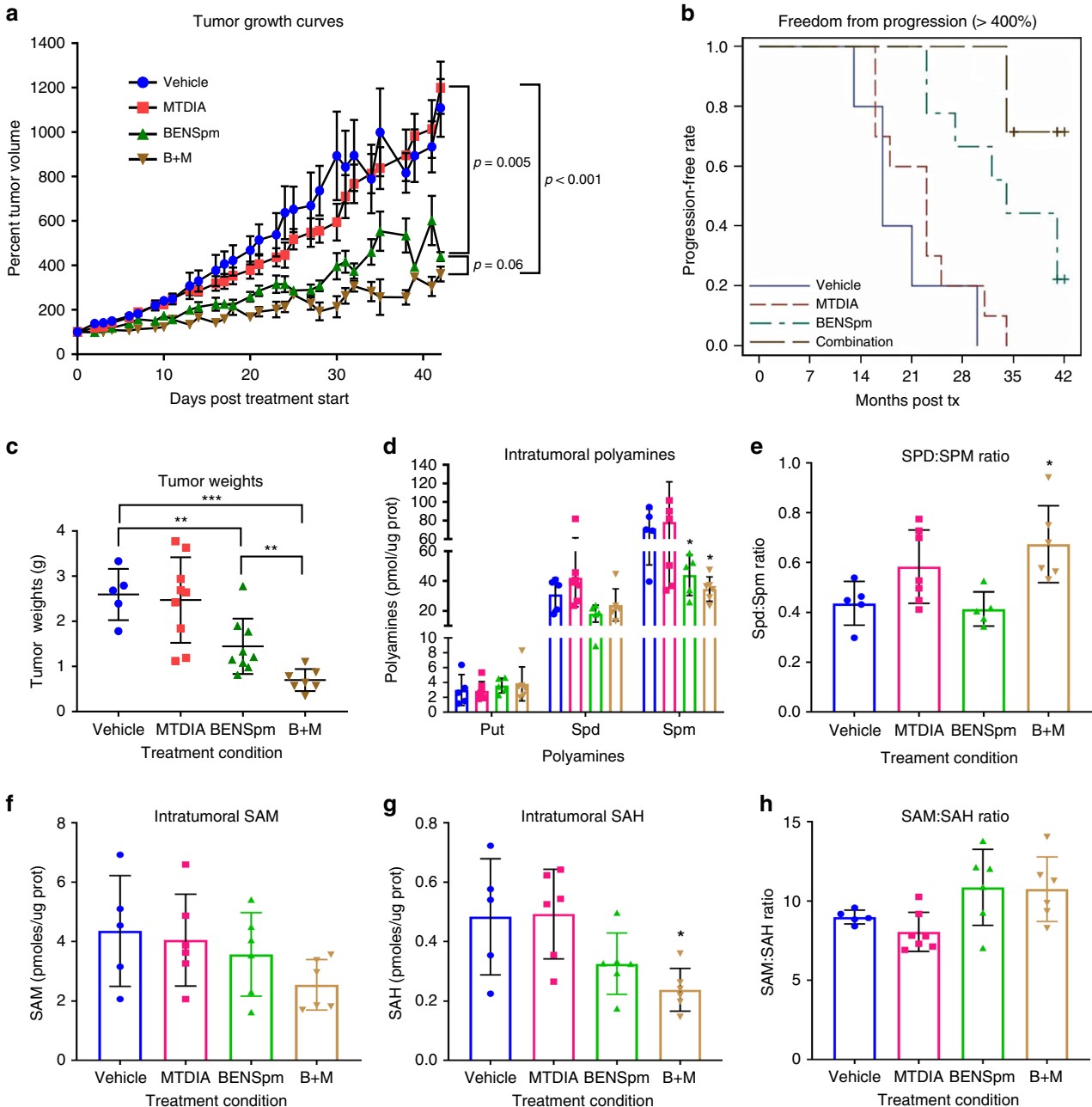

**Fig. 5 BENSpm and MTDIA in combination provide the greatest efficacy in vivo.** Animals were subcutaneously implanted with $1 \times 10^6$ CWR22Rv1 cells. Once tumors reached between 300 and 400 mm³ animals were placed on one of the four treatment cohorts, vehicle (blue), 50 mg/kg MTDIA (red), 100 mg/kg BENSpm (green) or the combination of 50 mg/kg MTDIA and 100 mg/kg BENSpm (brown). **a** Mean tumor sizes over time are plotted for each animal of the four treatment groups made relative to the starting tumor size at treatment day 0. The log relative tumor volume was modeled as a function of treatment, time, the time-treatment interaction, and random mouse effects and slopes using a linear mixed model. The growth rates were compared using tests about the appropriate linear contrasts of model estimates. (Vehicle; $n = 5$, MTDIA; $n = 11$, BENSpm; $n = 9$, Combo; $n = 8$). **b** The graph indicates progression free survival where progression is defined as reaching 400%. **c** The tumor weights at sacrifice for each group. (Vehicle; $n = 5$, MTDIA; $n = 9$, BENSpm; $n = 9$, Combo; $n = 7$). **d** Intratumoral polyamines (putrescine, spermidine, and spermine), **e** the spermidine-to-spermine ratio, **f** Intratumoral SAM levels, **g** Intratumoral SAH levels, and **h** the SAM:SAH ratio as measured by UPLC normalized to protein concentrations. (where for D-H Vehicle; $n = 7$, MTDIA; $n = 9$, BENSpm; $n = 9$, Combo; $n = 7$). Statistical analyses for graphs C-H were performed using an unpaired Student's t-test with Welch's correction. All values are compared to vehicle control unless otherwise indicated by connecting lines. Error bars represent standard deviation of the mean. *$p < 0.05$, **$p < 0.01$, ***$p < 0.001$.

calculation, we conclude that the drug combination has an additive effect in the in vivo setting.

Molecular analysis of intratumoral polyamine and SAM levels revealed that spermine levels were significantly decreased

following BENSpm and the combination treatment (Fig. 5d). Furthermore, the spermidine-to-spermine ratio was significantly increased following combination treatment (Fig. 5e), which suggested an upregulation of polyamine catabolism or

downregulation of polyamine synthase activity. SAM levels were decreased although not significantly (Fig. 5f). SAH levels were significantly decreased following combination treatment (Fig. 5g). There were no significant changes in the SAM-to-SAH ratio (Fig. 5h). The decrease in SAH following combination treatment suggested that flux from the methionine cycle to polyamine biosynthesis is being upregulated to support polyamine biosynthesis. Tissues were immunohistochemically (IHC) stained for 8-hydroxy-2′-deosyguanosine (8-oxo-dG), an indicator of oxidative DNA damage. 8-oxo-dG staining was significantly higher in BENSpm treated tumors, consistent with in vitro findings of increased ROS, which suggested that upregulation of polyamine catabolism resulted in increased ROS in vivo that was functionally significant (Supplementary Fig. 6A, B).

A second dosing schedule was tested for BENSpm that was given at 75 mg/kg i.p. once daily for 5 days, followed by 1 week off, giving three rounds of BENSpm treatment over the course of a 6-week study. BENSpm has a relatively long half-life and can remain at high levels in tissues for >4 days[8,17,38]. However, this schedule was not effective, which contrasts with the effectiveness of 100 mg/kg twice weekly schedule that resulted in decreased tumor growth rates, improved progression-free survival and decreased tumor weight at sacrifice (Supplementary Fig. 7A–C). This difference in efficacy of the two dosing regimens is of significant interest because previous clinical trials found little benefit when BENSpm was given once daily for a total of 5 days, and it has been suggested by members of the field that BENSpm given twice weekly for longer periods of time would limit toxicity and provide a greater overall benefit[17]. Therefore, these studies not only provide a combination therapy that is effective at blocking growth in vitro and in vivo, but also provide an optimal dosing regimen and schedule.

**BENSpm and MTDIA induce apoptosis in human samples.** An ex vivo explant system was used to evaluate the effects of the combination treatment in human CaP samples[39,40]. Fresh tissue defined as having >40% neoplastic involvement upon gross pathological review was collected from radical prostatectomies performed on treatment naïve patients at Roswell Park. Fresh tissue was placed on dental sponges in tissue culture dishes containing media with either vehicle control or the combination treatment (1 μM BENSpm + 1 nM MTDIA or 10 μM BENSpm + 10 nM MTDIA) for 7 days. Tissues defined as tumor were reviewed by a pathologist at Roswell Park who confirmed the presence of neoplastic cells by hematoxylin and eosin (H&E) staining of tissues following a 7-day treatment. Figure 6 includes representative images from 3 patient tumors treated with vehicle or the combination and stained for cleaved caspase-3 (Fig. 6a) or SSAT (Fig. 6b). CC3 was increased in 11 out of 19 treated tumor samples (Table 1), which indicated an increase in apoptosis following seven days of combination treatment. SSAT protein expression increased in 16 of 19 tumor samples (Table 1). The increased staining was observed in the epithelial cells surrounding the luminal structures (Fig. 6a) where prostate cancer arises and where polyamine production is high. Tumor polyamine levels were measured by UPLC (Table 1 and Fig. 6c–e). The drug combination led to reductions in tissue polyamines (spermidine + spermine) in 10 of 11 cases, as seen by a significant reduction in both spermidine and spermine levels compared to vehicle control. Furthermore, the change in SSAT protein expression was significantly and inversely correlated with the change in polyamine levels (spermidine + spermine) (Fig. 6f). Such correlation is expected since higher SSAT levels due to BENSpm treatment are predicted to deplete intracellular polyamine pools. Previous analysis using gene expression data from

The Cancer Genome Atlas (TCGA) prostate cancer cohort (PRAD) revealed that patients with dysregulation in the RNA expression of polyamine metabolic enzymes had shorter progression-free survival[41]. Others have shown that SSAT expression is high in patients who develop metastatic CaP[42]. Patients with tumors expressing high *SSAT* may be more susceptible to the combination therapy, based on the correlation in Fig. 6f. These findings indicated that a majority of androgen-sensitive patient samples tested with this ex vivo system were responsive to treatment suggesting the clinical potential for BENSpm and MTDIA.

**Discussion**
Previous work revealed that prostate cancer (CaP) is exquisitely sensitive to perturbation of polyamine metabolism and connected pathways due to high polyamine metabolic flux[9–11,13]. These studies demonstrated that the prostate relies heavily on the methionine salvage pathway to survive and that MTAP inhibition blocks LNCaP xenograft growth when given at the time of xenograft implantation[9]. In the current study, a combination strategy that utilizes MTAP inhibition was investigated. Rather than trying to work against the high degree of flux through polyamine metabolism that is inherent to prostate and CaP, flux was leveraged and enhanced to add metabolic stress. This approach relies on prostate cancer to maintain high polyamine biosynthesis at the expense of SAM depletion and ROS accumulation, rather than cutting off polyamine production. Unlike previous studies, the doses of BENSpm used in this study allowed for continued polyamine synthesis. Together, this added strain by increasing export via BENSpm treatment, while at the same time blocking the cells ability to mitigate that strain by inhibiting the methionine salvage pathway (via MTDIA).

It was shown that this combination was synergistic in vitro, blocked growth of established xenografts in vivo, and induced apoptosis in treated samples from patients that underwent radical prostatectomy. A differential molecular response to BENSpm and MTDIA was observed depending on AR status and androgen-independence in vitro. Interestingly, treatment depleted polyamine and SAM pools in androgen-sensitive lines but not androgen-independent lines, and androgen-sensitive lines upregulated export of acetylated polyamines to a greater extent. This suggested that cells may be unable to support flux where polyamine export is high. Furthermore, it may take longer for metabolite pools to be depleted in androgen-independent lines if enzymes and flux are not as high in this setting. The increased export of acetylated polyamines in the androgen sensitive setting, and therefore the downstream effects on SAM pools, is consistent with the observation of increased *SLC3A2* expression in these cell lines. *SLC3A2* is an acetylated polyamine exporter and the expression of this gene is enhanced by the androgen receptor.

Previous studies have attributed BENSpm's cytotoxic effects to *PAOX* and *SMOX* upregulation, both of which generate $H_2O_2$ and toxic aldehyde by-products during their enzymatic reactions[23]. In accordance with these findings, $H_2O_2$ accumulation was observed following treatment with BENSpm or the combination in prostate cancer cell lines, but to varying degrees. ROS accumulation and cytotoxicity to treatments was prevented by overexpressing a mitochondrial ROS clean-up enzyme (TXNRD2) in the androgen-independent lines, but not in the androgen-sensitive LNCaP line. This revealed that ROS accumulation is important for the cytotoxic effects seen in the androgen independent lines. Cells were treated with $H_2O_2$ to explore the inability of TXNRD2 overexpression to reduce ROS accumulation in LNCaP cells, and it was found that the threshold at which TXNRD2 overexpression could rescue ROS accumulation was higher in C4-2 cells. This is consistent with

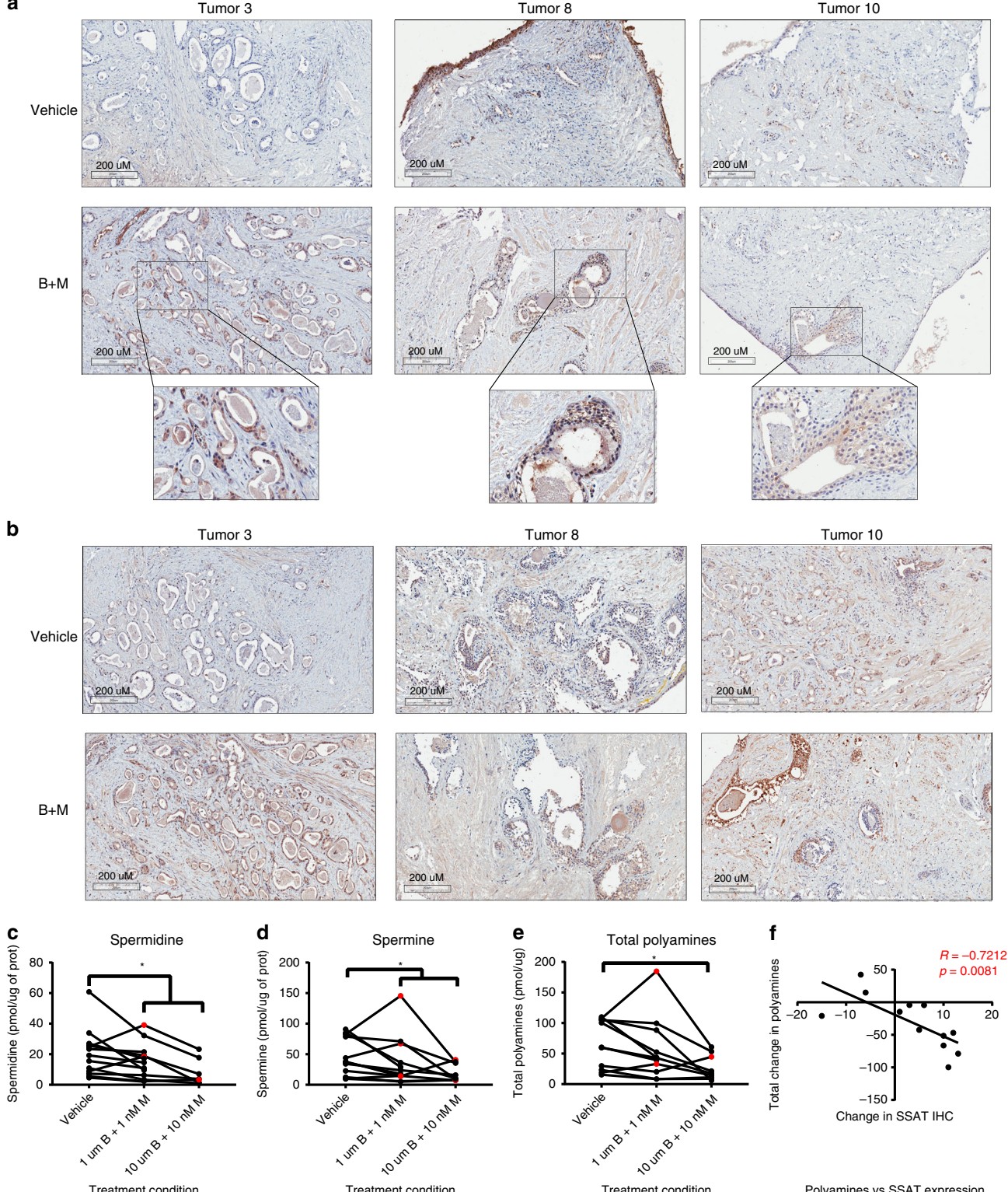

**Fig. 6 BENSpm and MTDIA impact patient samples in ex vivo system.** Ex vivo samples were collected from treatment naïve patients undergoing radical prostatectomy at Roswell Park Comprehensive Cancer Center. Tumors used for this study had at least 40% neoplastic involvement upon pathological review. Tumor samples were treated with vehicle control (vehicle), or combination treatment (1 nM MTDIA + 1 μM BENSpm or 10 nM MTDIA + 10 μM BENSpm) for 7 days in the presence of 20 μM MTA. Images from representative tumors 3, 8, and 10 from immunohistochemical stained slides for **a** cleaved caspase-3 (CC3) or **b** spermidine/spermine N1-acetyltransferase (SSAT) in both vehicle and the combination treated samples (B + M. Intratumoral (**c**) spermidine (Vehicle; n = 13, 1 + 1 BM; n = 13, and 10 + 10 BM; n = 7), **d** spermine (Vehicle; n = 12, 1 + 1 BM; n = 12, and 10 + 10 BM; n = 9), and **e** spermidine + spermine (Vehicle; n = 11, 1 + 1 BM; n = 11, and 10 + 10 BM; n = 9) levels normalized to protein concentrations for vehicle and combination (B + M) treated tumor samples. **f** The correlation between the change in SSAT IHC Score and the change in total spermidine + spermine levels measured by UPLC in the combination treated samples (n = 12). Pearson's r values are indicated. Statistical analysis for polyamines was done using a paired t-test. *p < 0.05.

**Table 1 Summary of all Patient Tumor Data. Indicated values for each parameter were calculated by subtracting combination treatment (using low dose for tumors 1 and 3, high dose for the remaining tumors) values from vehicle control for each tumor. Polyamine (spermidine + spermine) levels were measured by UPLC and normalized to protein concentration. Cleaved caspase-3 (CC3) and spermidine/spermine N1-acetyltransferase (SSAT) scores were determined based on the estimated percent positive cells multiplied by intensity of staining. The final column indicates the number of tumors that increased or decreased for each parameter over the total number of slides.**

| Tumor | 1 | 3 | 4 | 5 | 6 | 7A | 7B | 8 | 9 | 10 | 12 | 13 | 14 | 15 | 16 | 17 | 18 | 19 | 20 | Total |
|---|---|---|---|---|---|---|---|---|---|---|---|---|---|---|---|---|---|---|---|---|
| Polyamines | −20.9 | −66.4 | −78.9 | −42.5 | x | −51.4 | x | 14.8 | −14.5 | −4.3 | x | x | −4.4 | −46.9 | x | x | −99.7 | x | x | 10/11 ↓ |
| CCC | 0 | 9 | −5 | −9 | −2 | 6 | 10 | 2 | −1 | −3 | 15 | −2 | 5 | 2 | 19 | 8 | −2 | 13 | 8 | 11/19 ↑ |
| SSAT | −15 | 10 | 13 | 5 | 4 | 10 | 16 | −6 | 1 | 3 | 10 | 12 | 6 | 12 | 4 | 8 | 11 | −4 | 16 | 16/19 ↑ |

previous studies that showed C4-2 cells have a higher antioxidant capacity than LNCaP cells[34]. Though the *TXNRD2* overexpression experiment in LNCaP was inconclusive, it seems highly likely that increased ROS production in the LNCaP cell line contributes to the anti-proliferative effects of the treatment, which may combine with the metabolic effects that result in reduced intracellular polyamine levels and SAM:SAH ratio.

BENSpm and the combination treatment were shown to be effective in already established CWR22Rv1 xenografts grown in the castrate setting, with the greatest benefit seen with the combination. Furthermore, the dosing regimen for BENSpm of once daily for 5 days followed by 1 week off was not as effective as a higher dose given twice weekly. This is consistent with the suggestion that twice weekly continual therapies may be more efficacious in the clinic than once daily dosing done for 5 days[8]. Finally, treatments were effective at inducing apoptosis in a subset of androgen-stimulated patient samples ex vivo and induced changes in SSAT and MTAP protein expression as well as polyamine levels. Interestingly, not all patient tumors responded to treatments by increasing apoptosis and further work is needed to understand the molecular differences between patient samples. However, it is noteworthy that with 7 days of ex vivo treatment, 10 of 11 cases demonstrated a reduction in tissue polyamine levels relative to non-treated tissues in the same ex vivo conditions (Fig. 6c–e). In vitro studies indicated that most cell death is observed between 8 and 12 days of treatment. If the ex vivo cultures could be maintained for 12 days, a greater proportion of patient samples demonstrating increased apoptosis might be observed. The observation of reduced levels of polyamines in 10 of 11 cases, despite fewer cases exhibiting increased cleave caspase 3 is consistent with this idea.

This work provides an effective combination therapy that takes advantage of a metabolic vulnerability in CaP. Additionally, we demonstrated the potential of the ex vivo explant system, which can provide a simple platform to identify patients that are more likely to respond to treatment. Furthermore, this combination therapy can be combined with additional agents since no toxicities were seen in vivo. Docetaxel, a common agent used to treat mCRPC, disrupts microtubules which leads to increases in intracellular polyamine content[43]. Treatment with therapies that deplete polyamines released by chemotherapies can improve the overall therapeutic response. Therefore, docetaxel combined with BENSpm and MTDIA may be more efficacious in the clinic. This work also reveals a combination that may be useful in other cancer types in patients that do not have MTAP deletion. Pancreatic cancer, which generates a tremendous amount of spermidine and for which very few treatment options are available, is another cancer type where this combination approach might be efficacious. Additionally, recent studies have focused on identifying susceptibilities to loss of the methionine salvage pathway in MTAP deleted cancers[44–46]. The synergy reported here between

MTAP inhibition and activation of polyamine catabolism suggests that MTAP deleted tumors might also be sensitive to BENSpm treatment alone, and furthermore, that MTDIA might make cases that retain MTAP expression susceptible to therapeutic approaches designed to take advantage of MTAP loss.

These findings provide a compelling rationale for future studies exploring this combination strategy for treatment of various stages of prostate cancer progression. It was shown that the high rate of flux through polyamine biosynthesis in CaP can be leveraged in three ways: (1) dietary folate depletion was shown to be effective in models of both androgen-stimulated and castration recurrent CaP[11,13], (2) methionine salvage pathway inhibition was shown to be effective in androgen-sensitive models[9], and (3) the current study demonstrated that the combination of methionine salvage pathway inhibition and activation of polyamine catabolism was synergistic and highly effective in both androgen-sensitive and androgen-independent models of CaP. Both MTDIA and BENSpm have been used in numerous preclinical studies, with BENSpm being explored in solid tumors in clinical trials[25,26,41,47]. They have never been used in combination prior to the studies presented here, but they may provide the best clinical outcome when used alongside androgen deprivation therapy by synergizing with the effects of androgen withdrawal in CaP. Primary prostate cancer has highly dysregulated polyamine metabolic enzyme expression making it potentially susceptible to BENSpm and MTDIA[9,48]. Furthermore, these treatments do not rely on targeting of the androgen axis and may therefore work in combination with targeting of the androgen axis. Currently, there is a strong need for therapies which can reduce the rate of recurrence with low toxicity. Ultimately, leveraging this metabolic vulnerability in prostate cancer will provide therapies that are predicted to significantly delay or prevent disease recurrence.

## Methods

**Ultra-performance liquid chromatography for polyamines**. Ultra-Performance liquid chromatography analyses for polyamines were carried out similarly to previously described methods[10–13] with adjustments made to the flow rate, gradient, and column indicated below. Acetylated polyamines were extracted from media as previously described[10]. All polyamines measurements were carried out using an Acquity UPLC BEH Shield RP18 1.7 μm 2.1 × 100 mm column with a RP18 VanGuard Pre-column, 130 Å, 1.7 μm, 2.1 mm × 5 mm on an Acquity UPLC machine in the Bioanalytics, Metabolomics, and Pharmacokinetics Core Facility, at Roswell Park Comprehensive Cancer Center. The column was stored in 65% acetonitrile. A constant flow rate was held at 0.17 milliliters per minute, with a column temperature of 50 °C and sample temperature of 5 °C. Buffer A contained 55% 10 mM ammonium acetate at pH 4.4, and 45% HPLC grade acetonitrile. Buffer B was 100% acetonitrile. Dancylated polyamines were eluted with a linear gradient from 100% Buffer A to 18% Buffer A and 82% Buffer B for 6 min, which was then held for 3 min. By 10.6 min the conditions returned to 100% Buffer A, which also served to equilibrate the column for the next sample. Standards (Polyamines, acetylpolyamines and BENSpm) were also eluted using the same conditions and ranged in concentration from 1 to 100 μM. Spike-ins of each standard confirmed the location of the peaks within the samples. Concentrations of polyamines were determined based on standard curves and normalized to protein concentrations.

**Ultra-performance liquid chromatography for SAM and SAH**. Ultra-Performance liquid chromatography analyses for SAM and SAH were carried out with adjustments to previously described methods[10–13]. Alterations were made to the flow rate, gradient and column indicated below. In all, 650,000 cells were sonicated on ice in 55 μL or 10 mg of homogenized tissue was sonicated in 100 μL of 0.6 M PCA. Samples were centrifuged at 4 °C for 15 min at 10,000 × $g$, and the remaining supernatant was analyzed on the UPLC machine. Wherever possible, we utilized the same PCA extract for polyamines as we did for SAM and SAH analysis. All SAM and SAH measurements were carried out using an Acquity UPLC BEH Shield RP18 1.7 μm 2.1 × 100 mm column on an Acquity UPLC machine in the Bioanalytics, Metabolomics, and Pharmacokinetics Core Facility, at Roswell Park Comprehensive Cancer Center. Columns were stored in 65% acetonitrile. A constant flow rate was held at 0.17 milliliters per minute. Photodiode array detector measured absorbance at 250 nm. To make Buffer A 0.1 M NaH$_2$PO$_4$, 8 mM octane sulfonic acid, and 20 μM EDTA were combined brought to pH 2.65 with phosphoric acid and then filtered. Following filtration 2% acetonitrile was added to the buffer. For Buffer B the buffer was similar to Buffer A with two modifications; this buffer contains 0.15 M NaH$_2$PO$_4$ and was brought to pH 3.25. SAM and SAH were eluted with a linear gradient from 80% Buffer A to 0% Buffer A and 100% Buffer B for 4 min, which was then held for 2 min. By 6.8 min the conditions returned to 100% Buffer A, which also served to equilibrate the column for the next sample. SAM and SAH standards ranged in concentration from 0.01 nM to 10 μM. Spike-ins of either SAM or SAH standards confirmed retention time for peaks within samples. Concentrations of SAM and SAH were determined based on standard curves and normalized to protein concentrations.

**Immunohistochemistry**. Freshly harvested tissues were formalin fixed and paraffin embedded for immunohistological analysis as previously described[49] with the help of the Mouse Tumor Model Resource (MTMR) core at Roswell Park. Primary antibodies to Cleaved Caspase-3 from Cell Signalling (cat#: 9661), MTAP from Proteintech (Cat #: 11475–1-AP), SSAT from Santa Cruz (Cat #: sc-67159) and 8-oxo-dG from Biorbyt (Cat #: orb10011) were used at dilutions of 1:200, 1:100, 1:50, and 1:400 respectively. Following incubation with primary antibody, slides were incubated with a biotinylated goat anti-rabbit secondary antibody as previously described[49]. All CWR22Rv1 xenografts were analyzed by IHC for 8-oxo-dG and all ex vivo samples were analyzed by IHC staining of CC3, MTAP and SSAT. Analyses were carried out on the entire section with the percentage of positively stained cells counted manually and each section categorized as having <5%, 5–25%, 26–50%, 51–75%, or >75% positive cells. Analyses were carried out blind to treatment group.

**Western blotting**. Whole cell extracts were prepared and assayed as previously described[50]. Whole cell extracts were prepared in RIPA buffer (150 mM NaCl, 1.0% NP-40, 0.5% deoxycholate, 0.1% SDS, 50 mM Tris-HCl pH 8.0, 5 mM EDTA and 0.5 mM PMSF) supplemented with 1x proteinase inhibitor. Protein concentrations were determined by Pierce BCA Assay (Thermo Scientific) in technical triplicate. Samples were resolved on 12.5% polyacrylamide gels and transferred onto PVDF membranes (Biorad, Cat #162–0177). Blots were incubated with primary antibodies overnight at 4° and with secondaries for 1.5 h at room temperature. Samples were washed with 1x Tris-Buffered Saline (TBS) 3 times for 10 min each following primary and secondary antibody incubation. β-actin antibody[50] for Westerns was purchased from Sigma-Aldrich (Cat #A5441) and used at 1:2000. GAPDH[51] was from Santa Cruz Biotechnology Inc. (Cat #sc-25778) and used at 1:1000. SSAT [H-77] antibody was purchased from Santa Cruz Biotechnology (cat. # sc-67159) and used at 1:500. TXNRD2[33] antibody was purchased from Santa Cruz Biotechnology Inc. (Cat. # sc-46279) and used at 1:1000. SMOX and PAOX antibodies were a.pngt from the Casero lab (John Hopkins) and were both used at 1:1000. Secondary antibodies were purchased from Biorad (Cat. # 170–6515, 172–1011) and used at 1:2000. All antibodies were dissolved in 5% Biorad Blotting-Grade Blocker (Cat. #1706404). Signals were visualized using Pierce ECL western blotting substrate (Thermo Scientific, Cat #32209) and exposed using the Biorad ChemiDoc XRS. Band intensities were calculated using Image LabTM. Intensity values were normalized to β-actin loading control band intensities and made relative to control treatment conditions.

NOTE: Scramble control and shSSAT containing cells were treated for 96 h with 2.5 μM BENSpm to be able to detect SSAT protein expression by western blot. Uncropped western blots can be found in the Source Data file.

**RNA isolation and quantitative reverse transcriptase PCR**. RNA extractions were performed using standard TRIZOL extraction procedures and as previously described[50]. In all, 500 ng of RNA was retrotranscribed using the RevertAid First Strand cDNA Synthesis Kit (ThermoFisher Scientific) in a 20 μL reaction using a 1:1 mix of random hexamer primers and oligo DT, as per manufacturer's protocol. The cDNA was diluted 1:3. 1.5 μL of diluted cDNA was used for real-time reverse transcriptase PCR analyses, in duplicate, with the iTaq SYBR Green Supermix with ROX (Bio-Rad) on a StepOnePlusTM Real- Time PCR System (ThermoFisher Scientific). The reaction mixture contained 1.5 μL cDNA, 7.5 μL SYBR Green, 1.5 μL of 10 μM primer mix and 4.5 μL of water. The following primer sequences for *SSAT* (F-ATACTGCGGGCTGATCAAGGA and R-GCAAAACCAACAATG CTGTG), *GAPDH* (F-ACGGGAAGCTTGTGTCATCAAT and R-TGGACTCCACG

ACGTACTCA), *SLC3A2* (F-TCTTGATTGCGGGGACTAAC and R-GAGCCTTG CCTGAGACAAAC), and *GUSB* (F-CTCATTTGGAATTTTGCCGATT and R-CCGAGTGAAGATCCCCTTTTTA) were used.

**Cell culture conditions for combination treatments**. The androgen sensitive prostate cancer cell line LNCaP was purchased from the American Tissue Type Collection (ATCC, Manassas, CA). The androgen independent line LNCaP C4-2 and the androgen sensitive cell line LAPC-4 were kind gifts of Dr. James Mohler (RP). The androgen independent line CWR22Rv1 was a kind gift of Dr. Barbara Foster (RP). All cell lines were ATCC verified. LNCaP, LNCaP C4-2, and CWR22Rv1 were all maintained in RPMI 1640 with 10% FBS containing 1% penicillin streptomycin. LAPC-4 was maintained in RPMI 1640 with 15% FBS containing 1% penicillin streptomycin and 10 nM dihydrotestosterone (DHT). During treatments (described below) LNCaP and LAPC-4 cells were cultured in the same media they are maintained in. However, C4-2 and CWR22Rv1 cells were initially plated in their maintenance culture conditions but following 24 h to allow the cells to adhere, were washed with PBS and refreshed with RPMI 1640 media containing 2% charcoal strip FBS and 1% Penicillin/Streptomycin for 48 h prior to treatment start. This was done to simulate the androgen free castration environment for the androgen independent lines.

**In vitro combination treatments**. The MTAP inhibitor (3R,4S)-1-[(9-Deaza-adenin-9-yl)methyl]-3-hydroxy-4-(methylthiomethyl)-pyrrolidine, or MT-DADMe-Immucillin-A (MTDIA) was synthesized by Dr. Jim Phillips at the Cleveland Clinic Taussig Cancer Institute as previously described[9]. $N^1,N^{11}$-bis(ethyl) norspermine (BENSpm) was purchased at Synthesis Med Chem, Shanghai, China. Cells were plated at 200,000 cells per well in 6-well plates in their standard culture conditions described above. Following 24 h for LNCaP and LAPC-4 or 48 h of castration conditions for C4-2 and CWR22Rv1, cells were washed with PBS and refreshed with media containing vehicle control and 20 μM MTA, BENSpm alone, MTDIA alone or the combination of both at the indicated doses in the presence of 20 μM MTA for 8 days. The MTDIA doses ranged from 1 nM to 10 μM and the BENSpm doses ranged from 100 nM to 2.5 μM. Media and drug was refreshed every 48 h, and cells were trypsinized and plated in a larger vessel as needed. Cells were trypsinized following 8 days of treatment and counted by trypan blue exclusion. Experiments were performed in biological triplicate. Cell counts are represented as percent relative to untreated controls. Relative cell numbers for both individual drugs and the combination were calculated. Combination Indexes were calculated using the Chou Talaly method and CompuSyn program[29].

For mechanistic studies cells were treated as described above but using the synergistic drug combination of 1 nM MTDIA and 1 μM BENSpm. Analyses were performed for 8 days for LNCaP, LAPC-4, and C4-2 cells. For CWR22Rv1 cells mechanistic studies were performed for 96 h due to high cell death seen at 8 days. ROS analysis was performed at 8 days. All experiments were performed in biological triplicate unless otherwise indicated.

**Analysis of cell cycle progression by flow cytometry**. Cells were treated with Vehicle, 1 μM BENSpm, 1 nM MTDIA, or combination 1 μM BENSpm and 1 nM MTDIA for 8 days. For staining, 10x binding buffer was diluted 1:10 in distilled water to form a 1x binding buffer solution, on ice. Propidium iodide (PI) and fluorescein isothiocyanate (FITC)-conjugated annexin V (BD Biosciences) were utilized to detect cells undergoing apoptosis or necrosis. Cells were collected by trypsinization and centrifugation (1400 × $g$ for 4 min at 20 °C) after experimental treatments. Cells were then washed twice with cold phosphate-buffered saline (PBS). For staining, 10x binding buffer was diluted 1:10 in distilled water to form a 1x binding buffer solution, on ice. Cells were then resuspended in 1 × binding buffer at a concentration 1 × 10^6 cells/mL. Then, 100 μL of solution was transferred to a 5 mL flow cytometry tube and 5 μL of Annexin V-FITC and 5 μL of PI were added. No dye, FITC-annexin V only and PI only control samples were also generated. Cells were gently vortexed and incubated in the dark for 15 min at room temperature. Prior to flow-cytometric analysis, 400 μL of 1 × binding buffer was added, and cells were analyzed on a FACSCalibur™ flow cytometer (BD Biosciences, San Jose, USA) using forward-scattered light vs. side-scattered light as gating parameters. All analysis was conducted within an hour of initial staining procedure. FCS Express 6 was utilized for gating and flow cytometric analysis.

**MTAP activity**. MTAP activity was performed as previously described[22] with minor adjustments made indicated below. Treated cells were trypsinized, spun down, snap frozen in liquid nitrogen and kept at −80 °C until day of analysis. Cells were lysed in 0.6% Triton-X-100 buffer. A reaction mixture containing 50 μg of protein, 200 mM Na$_2$HPO$_4$, pH 7.56, 10 mM KCl, and 50 μM MTA containing 18,000 cpm [8-$^{14}$C] MTA. Radiolabeled MTA was purchased from Moravek Biochemicals, Inc (cat#: MC 185). Reactions were incubated at room temp for 8 min and stopped by addition of 0.6 M PCA. Reaction mixes were resolved on Silica thin layer chromatography paper in 1 mM Ammonium Acetate pH 7.55 containing 10% isopropanol. Adenine spots were excised and counted. All data is represented relative to control untreated cells.

**SSAT activity**. SSAT activity was assayed as previously described[52]. Treated cells were trypsinized, spun down, snap frozen in liquid nitrogen and kept at −80 °C until day of analysis. Cells were sonicated for 15 s on output 3 using the Fischer Scientific Sonic Dismembrator model 100 in 5 mM HEPES buffer. In all, 20 μL of extracted protein was combined with a reaction mixture containing 100 mM Bicine Buffer, 3 mM Spermidine and 0.01 mM [14C] Acetyl CoA. Radiolabeled acetyl CoA was purchased from Moravek Biochemicals, Inc (cat#: MC 269). Reactions were incubated at 37 °C for 5 min and stopped with the addition of hydroxylamine and boiled for 3 min. Completed reactions were spotted onto P81 Whatman filter paper, washed with running water for 10 min, placed in 100% Methanol, dried and then counted. Protein concentrations of extracted samples were determined using a Pierce BCA Protein Assay Kit (Thermo Fisher Scientific). Results are reported as specific activity relative to protein concentration (pmol/mg of protein/minute).

**ODC1 and AMD1 activity**. ODC and AMD1 activity was assayed as previously described[52]. Approximately 5 million cells were collected, washed with PBS and spun down. Cells were sonicated in breaking buffer containing DTT using a Fischer Scientific Sonic Dismembrator model 100 on output 3 for 3 rounds of 5 s on 5 s off, for a total of 15 s of sonication. Breaking buffer is made of 25 mM Tris/HCl at pH 7.5, 0.1 mM EDTA and 2.5 mM DTT. Following sonication samples were centrifuge at 4 °C for 10 min at 14,000 × g. The supernatant was removed and placed into a new, labeled tube. Depending on the enzyme of interest the following reaction mixtures were made up for either ODC or AMD1 activity for each sample. For ODC activity we combined 1 μL of 0.5 M Tris/HCl, 2 μL of 20 mM L-ornithine, 2 μL of 20 mM Pyridoxal-5 phosphate (B6), 5 μL of 25 mM DTT, 46 μL of ddH2O, and 4 μL of $^{14}$[CO$_2$]-Ornithine diluted 1:4. For AMD1 activity we combined 10 μL of 0.5 M Na$_2$PO$_4$, pH 7.5, 5 μL of 25 mM DTT, 5 μL of 4 mM S-adeno-sylmethionine, 20 μL of 15 mM putrescine, 16 μL of ddH2O, and 4 μL of $^{14}$[CO$_2$]-S-AdoMet diluted 1:1. After preparing the master mix for each reaction mixture the center well/stoppers were prepared. These are standard glass test tubes with rubber stoppers on top. Each center well was filled with 100 μL 1 M NaOH. Approximately 60 μL of reaction mixture prepared above was placed into the bottom of the test tube. In all, 40 μL of the sample extracted in breaking buffer was then quickly placed into the bottom of the test tube and samples were briefly vortexed and immediately capped with rubber stoppers containing center wells. Samples were incubated on a shaker for 1 h at 37 °C. After 1 h 0.5 mL of 5 M H$_2$SO$_4$ was injected through the rubber stopper past the center well into the reaction mix. Samples were again incubated on a shaker for 1 h at 37 °C. Scintillation vials were prepared containing 7.5 mL of scintillation fluid and 100 μL of 1 M HCl. After 1 h center wells were placed in scintillation vials, mixed and counted. A control sample of 40 μL breaking buffer and DTT (without protein) was also prepared that went through the same reaction procedure above. Protein concentrations were determined by Pierce BCA Assay in triplicate for each sample using the excess breaking buffer protein extract. Results are reported as specific activity relative to protein concentration (pmol/mg of protein/minute).

**SMOX and PAOX activity**. T-25 tissue culture flasks containing control and treated cells were washed with PBS and 1 mL of 0.0083 M glycine was added to the flasks. Flasks were then placed in −80 °C for at least 24 h. Flasks were thawed on ice and scraped. The lysates were placed in Eppendorf tubes and spun down at 4 °C, max speed for 15 min. The supernatant was then used for the determination of SMOX and PAOX activity. This procedure uses the ThermoFisher Amplex Red Kit. Using the 3% H$_2$O$_2$ standard provided, standards were prepared ranging from 1 to 10 μM. SMOX or PAOX reaction mixtures were prepared in a total volume of 55 μL. The reaction mixtures were prepared as follows with acetylated spermidine used for PAOX activity and spermine used for SMOX activity, 25 μL of 0.5 M glycine, 1.5 μL of 0.1 M aminoguanidine, 1.25 μL of 2.3 mg/mL pargyline, 22.25 μL of water, and 5 μL of 30 mM Spm/Ac-Spd. In all, 55 μL of reaction mix above was added to a 96-well white coated plate, along with 50 μL of lysate or 50 μL of standard, and 25 μL of 1x reaction buffer. The following reaction mix containing Amplex Red was then made up for each sample, 0.5 μL of 10 m Amplex Red, 1 μL of 10 U/mL HRP, and 48.5 μL of 1x Reaction Buffer. In all, 50 μL of the Amplex Red reaction mix was added to each well and the plates were covered with aluminum foil and incubated at room temperature for 1 h. After 1-h plates were read at 545 excitation and 590 nm emission on a 96-well plate reader. Protein concentrations were determined for lysates above. Standard curves were used to calculate H$_2$O$_2$ concentrations for each sample and then normalized to the protein concentrations and made relative to untreated controls.

**ROS Analysis**. 6-well plates containing control and treated cells were washed with PBS and then cells were scraped in 250 μL PBS. Lysates were placed in Eppendorf tubes and boiled at 90 °C for 10 min. Samples were then spun down at max speed for 5 min and allowed to cool on ice. This procedure also used is the Thermo Fischer Amplex Red Kit. However, for this 125 μL of PBS heated sample was added to a white coated plate along with 25 μL of 1x Reaction buffer. 50 μL of the Amplex Red reaction mix described above (containing Amplex Red, HRP and 1x Rxn Buffer) was then added to the plate, covered with aluminum foil and allowed to incubate at room temperature for 30 min. Plates were then read at 545 nm excitation and 590 nm emission on a 96-well plate reader. Fluorescence was normalized

to cell counts (described above) and all samples are represented as relative to control untreated cells.

**CWR22Rv1 xenograft studies**. Male Athymic Balb/c Nude mice were purchased from Envigo at ~2 months of age. In all, 48 mice were surgically castrated and injected with $1 \times 10^6$ CWR22Rv1 cells in matrigel subcutaneously into the right flank of castrated Nude mice. For each xenograft mice were randomly assigned to 1 out of 4 treatment groups; Vehicle Control, 100 mg/kg BENSpm, 50 mg/kg MTDIA, or 100 mg/kg BENSpm and 50 mg/kg MTDIA. Tumors were allowed to reach between 300 and 400 mm$^3$ at which point animals began treatment. Animals whose tumors were outside this range were excluded from the analysis. MTDIA was provided in the drinking water at 50 mg/kg. BENSpm was given intraperitoneal (i.p.) at 100 mg/kg twice weekly. Vehicle control mice received i.p. injections of water (BENSpm vehicle) two times per week and were given MTDIA vehicle in their drinking water. Animals receiving single agent therapies also received the other drug's vehicle. Tumor volumes were measured twice weekly using calipers and calculated using the following formula (width × width × length × 0.5234). Animals were monitored for signs of toxicity by twice weekly body weights, CBCs, and organ weights. No signs of toxicity were seen using these doses. Studies were completed following 6 weeks of treatment or once the tumors had reached greater than 2 cm$^3$ in diameter. All animal experiments were performed at Roswell Park Comprehensive Cancer Center animal housing facility in accordance with an Institutional Animal Care and Use Committee (IACUC) approved protocol.

**Ex vivo studies**. Fresh tissue from radical prostatectomies performed on treatment naïve patients at Roswell Park Comprehensive Cancer Center was collected for ex vivo analysis. Only tumors defined upon gross pathological review as containing greater than 40 percent neoplastic involvement were obtained. Matched adjacent non-tumor tissue was also collected. Fresh samples were cut up into approximately 10 milligram pieces in a tissue culture dish containing RPMI 1640 10% FBS and 1 % penicillin streptomycin. Gelatin Su.pngoam Sponges purchased from VWR (cat#: 10611–585) were placed in RPMI 1640 media containing 10% FBS, 1% Pen/Strep and 20 μM MTA with either vehicle control or 1 nM MTDIA and 1 μM BENSpm and allowed to incubate for 1 h. After 1 h, the tissue pieces were then placed on dental sponges in 6-well plates containing media with drug or vehicle. Treatments and media were refreshed every 48 h. Following 7 days of treatment tissues were placed in 10% formalin and paraffin embedded for IHC or snap frozen and kept at −80 °C for biochemical analysis. All of the tissue samples were collected under an Institutional Review Board (IRB)-approved protocol at Roswell Park Comprehensive Cancer Center. Specimens were collected after IRB-approved written consent from the patient was obtained at Roswell Park.

**Reporting summary**. Further information on research design is available in the Nature Research Reporting Summary linked to this article.

## Data availability
The source data underlying Supplementary Figs. 1C, 3A, and 4A are provided as a Source Data file. All the other data supporting the findings of this study are available within the article and its supplementary information files and from the corresponding author upon reasonable request. A reporting summary for this article is available as a Supplementary Information file.

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

## Acknowledgements

This work was supported in part by the following: DJS National Institute of Health R01CA197996; HCA National Institute of Health 1F99CA21245501; RAC National Institute of Health RO1CA204345, RO1CA235863. National Cancer Institute (NCI) grant P30CA016056 involving the use of Roswell Park Comprehensive Cancer Center's Mouse Tumor Model, Department of Laboratory Animal Research, Genomic, Biosta-tistics, and Bioanalytics, Metabolomics & Pharmacokinetics Shared Resources; The Roswell Park Alliance Foundation.

## Author contributions

H.C.A. and A.M.R. contributed to experimental design, writing, collecting and analyzing data, A.J.P., S.R.R., M.D.L., J.J.J., A.B.S. and C.S.B. contributed to writing, collecting, and analyzing data, B.M.G. and E.K. contributed to collecting data, B.A.F., M.M., J.H.W., K.A. and M.A.N. contributed to experimental design and collecting data, G.A. contributed to pathological analysis of samples and collecting of data, R.P., J.G.P., R.A.C. and D.J.S. contributing to experimental design, writing, collecting and analyzing data.

## Competing interests

The authors declare no competing interests.
