## [Peer Review File · Nature Communications]

Editorial note: Reviewer #3 was recruited in the second round of review to comment on the author's response to the original Reviewer #1 who was unavailable to review the revised manuscript.

Reviewers' comments:

Reviewer #1 (Remarks to the Author):

This work investigates how to take advantage of the specific upregulation of polyamine synthesis in prostate cancer for the treatment of this cancer type. Specifically, their data supports a synergistic or additive effect by the MTDIA+BENSpm drug combination targeting polyamine metabolism and methionine salvage. The reported in vitro data suggests a synergistic interaction, although additional evidence is required (see below). The in vivo data suggests at least an additive effect without additional toxicity. The work is very interesting and it makes use of different analytical techniques to dissect the drug mechanism of action. The reported in vivo data indicates the MTDIA+BENSpm as a promising drug combination for the treatment of prostate cancer.

There are some points in the manuscript that require further analysis to support the authors' conclusions:

Line 110: The authors report that "FACS sorting of Annexin-V and Propidium Iodide stained LAPC-4 cells revealed a decreased number of live cells, an increased number of early and late apoptotic cells, and no changes in the number of necrotic cells with treatment (Fig. 2E)". However, the data reported in Fig. 2E is not sufficient to arrive to that conclusion. The authors should report (e.g., in bar plots) the specific quantifications of the number of early apoptotic cells, late apoptotic cells, and necrotic cells; including the average and standard deviation across 3 or more biological replicates and the statistical significance of the changes relative to untreated controls.

Line 115: The authors conclude that "These experiments proved that BENSpm and MTDIA treatment was synergistic and cytotoxic following 8 days of treatment". I agree that from the analysis of the Chou-Talalay combination index the authors can conclude that there is a synergistic interaction between the growth inhibition by MTDIA+BENSpm. However, from the data reported it cannot be concluded that there is a synergistic interaction with respect to induction of apoptosis. The authors should present evidence that the induction of apoptosis by the combination MTDIA+BENSpm is significantly higher than what expected from the additive induction of apoptosis by MTDIA and BENSpm when used as single drugs. For example, the authors could use the following methodology. Let's denote by p_1 and p_2 the fraction of apoptotic cells when treating with drug 1 and 2, respectively. In the absence of drug interaction, the fraction of cells that do not engage in apoptosis because of the action of neither drug is $(1-p_1)(1-p_2)$. Consequently, the expected fraction of apoptotic cells due to the independent action of both drugs is $p_{12} = 1 - (1-p_1)(1-p_2)$. This expected value in the absence of interaction can be compared to the observed value P_{12} . If P_{12} is significantly higher than p_{12} then we have evidence that there is a synergistic interaction regarding the induction of apoptosis. For the assessment of the statistical significance, the authors can use data for 3 biological replicates for p_1 , p_2 and P_{12} , generate the 9 possible values of p_{12} , and perform a t-test between the values of P_{12} and p_{12} .

Line 118: The authors report that "Cells were treated for 8 days in the presence of 20 μ M MTA with 1 nM MTDIA and 1 μ M BENSpm (a synergistic dose in LNCaP, C4-2 and CWR22RV1) for the remainder of the mechanistic studies". The authors should either present data that the interaction is synergistic with respect to apoptosis (see comment above) or report that the selected dose is synergistic only with respect to growth inhibition.

Line 180: The authors report that H₂O₂ levels are increased upon BENSpm treatment (Fig. 4C), which is consistent with the reported increase of PAOX and SMOX activity (Fig. 4B). However, this association is not sufficient for a causal relationship. The authors should knockdown (e.g., by siRNA) the expression of PAOX and SMOX and measure H₂O₂ levels relative to control. This data is important to establish the mechanistic connection between BENSpm and H₂O₂ elevation. This should be done in the 3 cell lines LNCaP, C4-2 and CWR22Rv1. Since three independent cell lines would be tested, one biological replicate per cell line would be sufficient.

Line 233: The authors report “Strikingly, the combination treatment was more effective than either drug alone as evident by the tumor weights at sacrifice (Fig. 5C). These in vivo findings are in agreement with the synergistic effect of the combination identified in vitro (Fig. 2A).” This statement would suggest that the combination is synergistic in the in vivo setting. However, the reported data is not sufficient to arrive to this conclusion. By this I do not mean to diminish the importance of the observation. In my opinion, showing evidence for an additive effect with regard to growth inhibition, without an increase in toxicity, is sufficient to consider the combination promising. At the same time, it would be desirable to know whether the tumour growth inhibition is additive or, even better, synergistic. As an optional suggestion, the authors could use the methodology described above to discriminate between an additive or synergistic effect. Otherwise, they could tune down the wording and report that, regarding tumor growth inhibition in vivo, there is at least an additive effect.

Reviewer #2 (Remarks to the Author):

In the manuscript “Drug-Induced Increases in Polyamine Catabolism Synergize with Inhibition of the Methionine Salvage Pathway: A Novel Approach for Prostate Cancer Therapy”, authors exploit the requirement of polyamine for prostate cancer cell proliferation. They show that enhanced polyamine catabolism (by increasing spermidine/spermine N1-acetyltransferase or SSAT; achieved by treatment with BENSpm) generates a metabolic stress which is mitigated by prostate cancer cells by enhancing the methionine salvage pathway that replenishes polyamines. They further suggest that enhancing the levels of SSAT activity combined with blockage of methionine salvage pathway (using an inhibitor of methyladenosine phosphorylase; MTAP inhibitor MTD compound) could synergise cell killing to achieve effective tumour growth inhibition. These findings are not novel as similar observations have been made by this group earlier (see point 1 below).

1. Specifically, the authors have earlier shown that increasing SSAT activity sensitises prostate tumor cells to folate depletion, which feeds into methionine salvage pathway as exploited by this group earlier (FASEB J. 23(9): 2888-2897). The current study essentially exploits their earlier FASEB J study.

2. In the earlier study above, authors found an additive effect of activating SSAT coupled with folate depletion. However, the current study shows synergism. In the discussion section, an explanation for these different observations should be provided.

3. Authors quoted earlier research (Basu et al. PMID: 19773450) that suggests a role for polyamine oxidase (PAOX) in replenishing polyamines. This work shows inhibiting polyamines synthesis pathway can result in inhibition of cell growth. My query is if blocking this pathway alone has a drastic impact on growth inhibition, why does this present study suggest that PAOX is involved only to a “lesser extent”.

4. Basu et al (above study) had earlier shown that SSAT activity can be induced by androgens, in which case should combination of androgens to induce SSAT activity with MTAP inhibition lead to similar observations as shown here? The fact that majority of the prostate cancer cells have heightened androgen signalling it is natural that blocking MTAP alone should lead to similar phenotypic consequences as shown here.

5. Figure 3B: authors suggest that treatment of prostate cancer cell lines by SSAT inducer (BENSpm) and its combination with MTAP inhibitor MTD enhanced polyamine catabolism (levels of secreted acetylated polyamines). It is not clear why MTAP inhibition by MTD alone increases polyamine secretion in LNCaP and CWR22Rv1 cells. Moreover, as opposed to what authors suggest

the combination treatment does not appear to enhance the secreted polyamine levels in CWR22Rv1 cells compared to BENSpm alone. Given that the growth of CWR22Rv1 cells is highly sensitive to the combination (see figure 2C) one would expect that these cells will show enhanced polyamine catabolism as a result of combinatorial treatment.

6. C4-2 cells are derived from LNCaP but are highly aggressive. Authors should explain the rationale underlying increased SSAT activity in LNCaP cells in control conditions, which is diminished in C4-2 cells (supplementary figure 1C).

We thank the reviewers for their thoughtful comments. Below we give a point by point response:

Reviewer 1 Comments:

1. Line 110: The authors report that “FACS sorting of Annexin-V and Propidium Iodide stained LAPC-4 cells revealed a decreased number of live cells, an increased number of early and late apoptotic cells, and no changes in the number of necrotic cells with treatment (Fig. 2E”. However, the data reported in Fig. 2E is not sufficient to arrive to that conclusion. The authors should report (e.g., in bar plots) the specific quantifications of the number of early apoptotic cells, late apoptotic cells, and necrotic cells; including the average and standard deviation across 3 or more biological replicates and the statistical significance of the changes relative to untreated controls.

 - Thank you for your suggestion, we have incorporated these changes and added quantification for data in Figure 2E, which strengthens the finding that treatment of LAPC4 significantly decreases the number of live cells and increases the number of cells in early and late apoptosis, with small but statistically significant changes in necrosis. We have additionally plotted these data for the other 3 cell lines.
2. Line 115: The authors conclude that “These experiments proved that BENSpm and MTDIA treatment was synergistic and cytotoxic following 8 days of treatment”. I agree that from the analysis of the Chou-Talalay combination index the authors can conclude that there is a synergistic interaction between the growth inhibition by MTDIA+BENSpm. However, from the data reported it cannot be concluded that there is a synergistic interaction with respect to induction of apoptosis. The authors should present evidence that the induction of apoptosis by the combination MTDIA+BENSpm is significantly higher than what expected from the additive induction of apoptosis by MTDIA and BENSpm when used as single drugs. For example, the authors could use the following methodology. Let’s denote by p_1 and p_2 the fraction of apoptotic cells when treating with drug 1 and 2, respectively. In the absence of drug interaction, the fraction of cells that do not engage in apoptosis because of the action of neither drug is $(1-p_1) \times (1-p_2)$. Consequently, the expected fraction of apoptotic cells due to the independent action of both drugs is $p_{12} = 1 - (1-p_1) \times (1-p_2)$. This expected value in the absence of interaction can be compared to the observed value P_{12} . If P_{12} is significantly higher than p_{12} then we have evidence that there is a synergistic interaction regarding the induction of apoptosis. For the assessment of the statistical significance, the authors can use data for 3 biological replicates for p_1 , p_2 and P_{12} , generate the 9 possible values of p_{12} , and perform a t-test between the values of P_{12} and p_{12} .

 - We used the above method, as suggested, to determine a theoretical P_{12} (combination of BENSpm and MTDIA treatment) where p_1 indicated BENSpm treatment and p_2 indicated MTDIA treatment. We then compared the theoretical p_{12} output with the actual combination values. Based on a t-test, there was a statistically significant difference ($p=0.0098$) between the two; however, this was not a synergistic difference, but rather an additive one. Therefore, in regards to the flow cytometric data, we have removed the use of the word “synergistic”.
3. Line 118: The authors report that “Cells were treated for 8 days in the presence of 20 μ M MTA with 1 nM MTDIA and 1 μ M BENSpm (a synergistic dose in LNCaP, C4-2 and CWR22RV1) for the remainder of the mechanistic studies”. The authors should either present data that the

interaction is synergistic with respect to apoptosis (see comment above) or report that the selected dose is synergistic only with respect to growth inhibition.

- Due to the results reported in the previous reviewer comment, we have removed the word “synergistic” in reference to the apoptotic flow cytometric data and will only be using it in reference to the growth inhibition data, for which we show a synergistic combination index. We changed the text to read:
“a synergistic dose with respect to growth inhibition in LNCaP, C4-2 and CWR22RV1)”
- 4. Line 180: The authors report that H₂O₂ levels are increased upon BENSpm treatment (Fig. 4C), which is consistent with the reported increase of PAOX and SMOX activity (Fig. 4B). However, this association is not sufficient for a causal relationship. The authors should knockdown (e.g., by siRNA) the expression of PAOX and SMOX and measure H₂O₂ levels relative to control. This data is important to establish the mechanistic connection between BENSpm and H₂O₂ elevation. This should be done in the 3 cell lines LNCaP, C4-2 and CWR22Rv1. Since three independent cell lines would be tested, one biological replicate per cell line would be sufficient.
- We appreciate the suggestion, and thus have knocked down PAOX and SMOX via shRNAs (one non-silencing (NS) and two targeting shRNAs per gene) to assess the causal relationship between PAOX/SMOX activity and ROS induction. This is included as supplemental figure 3. Both shRNAs for each gene significantly reduced the protein level of SMOX/PAOX in LNCaP cells. We then treated these LNCaP cells (NS-control, shSMOX/shPAOX) for 8 days with either vehicle control, 1nM MTDIA, 1uM BENSpm, or the combination treatment and measured ROS levels (as in figure 4). We found that knockdown of SMOX was able to reduce both basal levels of ROS and treatment-induced ROS in LNCaP cells. This is consistent with a reduction in SMOX activity after 8 days in at least one of the shSMOX cell lines compared to NS-Control. Furthermore, we found that PAOX knockdown cells had ROS levels comparable to NS-control cells in basal and treatment-induced conditions cells. This is in agreement with the PAOX enzyme activity of the shPAOX lines having similar PAOX activity to the NS-control cells after 8 days of treatment with either vehicle control, 1uM BENSpm, 1nM MTDIA, or B+M, despite having reduced PAOX protein expression. Therefore, the increased ROS observed in LNCaP cells upon treatment with BENSpm or combination is due to induction of SMOX enzyme activity, since knockdown of SMOX is able to rescue levels of BENSpm- and combination-induced ROS. Therefore, we believe SMOX is the main contributor of ROS induced by treatment.
- In addition, we used the same sets of shRNAs in C4-2 and RV1 cells. We initially screened a panel of 5 shRNAs for each gene and assessed by RT-PCR which ones worked best at reducing mRNA. The two shRNAs chosen for each gene were effective in reducing the mRNA in all three cell lines. Despite the reduction in mRNA, in the C4-2 cells and the CWR22RV1 cells, we had some unexpected findings. For example, when we knocked down SMOX in C4-2 cells, we found that even though the mRNA was reduced >75%, protein levels were maintained, as was SMOX activity. Similarly, despite a >50% knockdown of PAOX mRNA, PAOX protein levels were not reduced and neither was activity. In addition, we were unable to maintain the CWR22RV1 line with one of the SMOX shRNAs, but with the one we were able to maintain we found >50% mRNA decrease and a similar decrease in protein level, however no loss in SMOX activity. These experiments were repeated multiple times to confirm the results, and the shRNAs being expressed were confirmed by sequencing. These findings suggest that there are

multiple levels of compensatory regulation for SMOX and PAOX that are too complicated to sort out for this paper. Therefore, we were unable to address the question in these cell lines. These findings implicate complex compensatory mechanisms in these cell lines.

5. Line 233: The authors report “Strikingly, the combination treatment was more effective than either drug alone as evident by the tumor weights at sacrifice (Fig. 5C). These in vivo findings are in agreement with the synergistic effect of the combination identified in vitro (Fig. 2A).” This statement would suggest that the combination is synergistic in the in vivo setting. However, the reported data is not sufficient to arrive to this conclusion. By this I do not mean to diminish the importance of the observation. In my opinion, showing evidence for an additive effect with regard to growth inhibition, without an increase in toxicity, is sufficient to consider the combination promising. At the same time, it would be desirable to know whether the tumour growth inhibition is additive or, even better, synergistic. As an optional suggestion, the authors could use the methodology described above to discriminate between an additive or synergistic effect. Otherwise, they could tune down the wording and report that, regarding tumor growth inhibition in vivo, there is at least an additive effect.
- The *Bliss Combination Index* approach shows that a positive drug combination effect occurs when the observed combination effect (E_{AB}) is greater than the expected additive effect given by the sum of the individual effects ($E_A + E_B$). The Combination Index can then be calculated as: $CI = \frac{E_A + E_B}{E_{AB}}$. A corresponding *P*-value is then given by the significance of the interaction effect in a factorial analysis of variance of the individual and combination effects.

When the Bliss combination method was applied to in Vivo experiments for tumor weights and Tumor Growth Inhibition Rates, separately (where A indicated the BENSpm treatment, B indicated the MTDIA treatment, and AB indicated the Combination Treatment), an interaction index was produced. Synergistic effects were denoted where AB was greater than A+B, Additive was denoted where AB = A+ B. We found that for both measures of tumor weight and of growth rate the combination effect was additive. Therefore, we have removed the word “synergistic” regarding the combination approach from the in vivo data; however, it is still important to note that via dose response curves and the Chou-Talalay method, 1 μ M BENSpm and 1 nM MTDIA were found to be highly synergistic in multiple cell lines. We added the following statement:

“We applied the Bliss Combination Index (37) approach for both tumor weight and growth inhibition rates to ask if the combination therapy was additive or synergistic. Based on this calculation, we conclude that the drug combination has an additive effect in the in vivo setting.”

Reviewer 2 Comments:

1. 1. Specifically, the authors have earlier shown that increasing SSAT activity sensitizes prostate tumor cells to folate depletion, which feeds into methionine salvage pathway as exploited by this group earlier (FASEB J. 23(9): 2888-2897). The current study essentially exploits their earlier FASEB J study.

- While the current study builds on the concept of leveraging the prostate's unique sensitivity to manipulation of the polyamine and connected metabolic pathways (as above), the approaches and effects are vastly different. The current study utilizes MTDIA, a pharmacological inhibitor of the methionine salvage pathway targeting the rate limiting enzyme methylthioadenosyl phosphorylase (MTAP), which has direct effects on SAM recycling for use in the polyamine pathway. Furthermore, the addition of the polyamine analogue, Bisethyl norspermine (BENSpm) we are able to enhance the inherently high activity of SSAT and thus flux through the polyamine pathway specifically in the prostate simply by stabilizing the protein. Rather than limiting resources such as SAM for polyamine production through dietary folate depletion, BENSpm and MTDIA together induce metabolic stress capable of activating catabolism and inducing ROS and apoptosis. The targets of each approach occur at different points of one carbon metabolism, and thus can affect many different aspects of metabolic outputs.
2. In the earlier study above, authors found an additive effect of activating SSAT coupled with folate depletion. However, the current study shows synergism. In the discussion section, an explanation for these different observations should be provided.
- In the earlier study, we did not test for synergy. In that study we used experimentally manipulated the levels of folate in the media in combination with a genetically based overexpression of SSAT. In contrast, here we are using two pharmacological agents, neither of which directly affects folate. Given these differences, we find no expectation that the results should be the same. This again speaks to the novelty of the current work with respect to the earlier study.
3. Authors quoted earlier research (Basu et al. PMID: 19773450) that suggests a role for polyamine oxidase (PAOX) in replenishing polyamines. This work shows inhibiting polyamines synthesis pathway can result in inhibition of cell growth. My query is if blocking this pathway alone has a drastic impact on growth inhibition, why does this present study suggest that PAOX is involved only to a "lesser extent".
 - We agree with the reviewer that our characterization of PAOX being involved "to a lesser extent" was inaccurate. Therefore, in the introduction, we removed this phrase.
4. Basu et al (above study) had earlier shown that SSAT activity can be induced by androgens, in which case should combination of androgens to induce SSAT activity with MTAP inhibition lead to similar observations as shown here? The fact that majority of the prostate cancer cells have heightened androgen signaling it is natural that blocking MTAP alone should lead to similar phenotypic consequences as shown here.
- While androgens have been shown to transcriptionally regulate key enzymes of the polyamine pathway, the extent to which they upregulate SSAT is not comparable to the effect that BENSpm has. BENSpm, unlike androgens, binds specifically to SSAT and activates its activity more than 100-fold. Therefore, the impact of BENSpm on SSAT is more specific and has greater induction of its target enzyme. In our experience, the addition of the androgen dihydrotestosterone slightly increases the effect of BENSpm on SSAT activity, but the SSAT induction is not dependent on it (data not shown). Therefore, androgen sensitive cells may be more sensitive to MTAP inhibition, but the effects of BENSpm with MTAP inhibition are much more specific.

5. Figure 3B: authors suggest that treatment of prostate cancer cell lines by SSAT inducer (BENSpm) and its combination with MTAP inhibitor MTD enhanced polyamine catabolism (levels of secreted acetylated polyamines). It is not clear why MTAP inhibition by MTD alone increases polyamine secretion in LNCaP and CWR22Rv1 cells. Moreover, as opposed to what authors suggest the combination treatment does not appear to enhance the secreted polyamine levels in CWR22Rv1 cells compared to BENSpm alone. Given that the growth of CWR22Rv1 cells is highly sensitive to the combination (see figure 2C) one would expect that these cells will show enhanced polyamine catabolism as a result of combinatorial treatment.
 - We do not have a certain explanation for why MTAP inhibition with MTDIA leads to increased acetylated polyamine secretion in CWR22Rv1 (the increase is not significant in LNCaP). One possibility is that the increase in MTA caused by MTAP inhibition may block the activity of polyamines synthases. This may activate catabolism under low synthesis activity, resulting in increased secreted polyamines. In regards to RV1 cells, combination treatment does not increase acetylated polyamines more so than BENSpm alone. Since RV1 cells are most sensitive to the combination treatment, the lack of induction in acetylated polyamines is likely due to increases cell death observed in cells treated with combination for 8 days.

6. C4-2 cells are derived from LNCaP but are highly aggressive. Authors should explain the rationale underlying increased SSAT activity in LNCaP cells in control conditions, which is diminished in C4-2 cells (supplementary figure 1C).
 - The reduced SSAT expression in C4-2 cells likely due to the fact that C4-2 is an androgen insensitive line and so the levels of SSAT induction are less than LNCaP. Though the SSAT activity is slightly lower as well compared to LNCaP (Fig 2.), BENSpm highly induces SSAT activity comparable to in LNCaP.

Reviewers' comments:

Reviewer #2 (Remarks to the Author):

The revised manuscript has satisfyingly answered some but not all of the critiques. As for point 6, authors should note that with regards to the original report from Thalmann et al., 2000 in "Prostate" the C4-2 were initially deemed androgen-independent. We now know their growth can be stimulated by androgens and more immortally repressed by new generation anti-androgens such as Enzalutamide. Therefore, it is not appropriate to call them androgen insensitive. The combination of (A) Emphasis/de-emphasis on some of their original points such as necrosis/synergy, (B) their earlier report titled "Polyamine biosynthesis impacts cellular folate requirements necessary to maintain S-adenosylmethionine and nucleotide pools" in FASEB J 23(9):2888 providing this concept exploited in this manuscript, (C) lack of mechanism underlying increased ac-polyamine secretion in response to MTAP inhibition, make it unclear whether this report contains enough novel information to merit publication in Nature Communications.

Reviewer #3 (Remarks to the Author):

In this revised manuscript, Affronti et al. addressed the majority of the concerns raised by the referees. In particular, beyond some technical issues that the authors addressed appropriately, referee 1 asked whether the combination therapy with BENSpm and MTDIA is synergistic with respect to apoptosis or growth inhibition in vitro and in vivo. This request prompted the authors to perform additional analyses that revealed an additive of the drugs with regards to apoptosis, and a synergistic one with regards to proliferation. Also, they found that the drug combination has an additive effect in the in vivo setting. The authors also added new functional analyses to investigate the impact of the silencing of SMOX and PAOX on drug treatment. These experiments helped to clarify the supposed mechanism of action of the drugs tested. Overall, these new data added strength to the authors' conclusions and adequately addressed the referee's concerns.

We thank the reviewers for their thoughtful comments. Below we give a point by point response:

Reviewer 3 Comments:

1. In this revised manuscript, Affronti et al. addressed the majority of the concerns raised by the referees. In particular, beyond some technical issues that the authors addressed appropriately, referee 1 asked whether the combination therapy with BENSpm and MTDIA is synergistic with respect to apoptosis or growth inhibition in vitro and in vivo. This request prompted the authors to perform additional analyses that revealed an additive of the drugs with regards to apoptosis, and a synergistic one with regards to proliferation. Also, they found that the drug combination has an additive effect in the in vivo setting. The authors also added new functional analyses to investigate the impact of the silencing of SMOX and PAOX on drug treatment. These experiments helped to clarify the supposed mechanism of action of the drugs tested. Overall, these new data added strength to the authors' conclusions and adequately addressed the referee's concerns.
- We appreciate the effort of the 3rd reviewer to assess our response to reviewers 1 and 2 and that the review found that we adequately addressed reviewer 1 and 2 concerns. We further note that reviewer 3 found that our added experiments help to clarify the mechanism of action of the drugs tested.

Reviewer 2 Comments:

1. The revised manuscript has satisfyingly answered some but not all of the critiques. As for point 6, authors should note that with regards to the original report from Thalmann et al., 2000 in "Prostate" the C4-2 were initially deemed androgen-independent. We now know their growth can be stimulated by androgens and more immortally repressed by new generation anti-androgens such as Enzalutamide. Therefore, it is not appropriate to call them androgen insensitive.
- In response to the original review from reviewer #2, we addressed the question of why SSAT levels were higher in LNCaP cells than in C4-2 cells (supplemental figure 1c) which are derived from LNCaP but are highly aggressive. In that response, we were imprecise in our language and referred to C4-2 cells as "androgen insensitive". They key point, however, is that we performed our experiments in C4-2 cells in androgen-free conditions (phenol red-free RPMI 1640 media containing 2% charcoal strip FBS). Androgens can upregulate the expression of SSAT. The LNCaP experiments are not done in androgen free conditions. Therefore, we explain the lower level of SSAT in C4-2 cells, as compared to LNCaP cells, to be due to the androgen free culture conditions. SSAT activity is also slightly lower in C4-2 cell compared to LNCaP (Fig 2.). Nevertheless, BENSpm highly induces C4-2's SSAT activity in androgen free conditions to a level that is comparable to LNCaP. This and many other experiments in the paper, particularly the CWR22Rv1 xenograft experiments performed in castrate male nude mice, indicate that our novel therapeutic approach can be effective in the castrate environment. This is of significance because the major clinical challenge in managing prostate cancer is recurrence during androgen deprivation therapy. Our studies demonstrate that the therapeutic approach we are using is not dependent on the

androgen axis, which is being targeted in androgen deprivation therapy, and therefore might be able to be used alongside androgen deprivation therapy as a means to prevent or delay recurrence.

2. The combination of **(A)** Emphasis/de-emphasis on some of their original points such as necrosis/synergy, **(B)** their earlier report titled “Polyamine biosynthesis impacts cellular folate requirements necessary to maintain S-adenosylmethionine and nucleotide pools” in FASEB J 23(9):2888 providing this concept exploited in this manuscript, **(C)** lack of mechanism underlying increased ac-polyamine secretion in response to MTAP inhibition, make it unclear whether this report contains enough novel information to merit publication in Nature Communications.
- For the first part of this statement concerning synergy, we found that the drug combination was synergistic in terms of proliferation, but additive in terms of inducing apoptosis, and additive in the in vivo setting. We do not agree that the fact that we see additive effects rather than synergistic detracts from the novelty of this work. In fact, when reviewer #1 questioned whether the in vivo effects were additive or synergistic, the reviewer specifically made the point that, *“By this I do not mean to diminish the importance of the observation. In my opinion, showing evidence for an additive effect with regard to growth inhibition, without an increase in toxicity, is sufficient to consider the combination promising.”*
 - For the second part of the statement, the reviewer is asserting that our work is not sufficiently novel because our 2009 FASEB Journal article already provided the concept. We believe this is an inaccurate characterization of the novelty of the current manuscript for the following reasons.

The 2009 manuscript deals strictly with the concept that prostate cancer cell lines are more dependent upon folic acid in the media than colorectal cancer cell lines, and this is due to their higher level of polyamine biosynthesis. We demonstrate the following additional three points in that article:

1. Limiting folic acid in the media can combine with over expression of SSAT in a single cell line to enhance the antiproliferative effects and the reduction of the s-adenosyl methionine (SAM) pools. This experiment served to strengthen the case that there is a relationship between polyamine metabolism and the enhanced dependence on folic acid.
2. Blocking polyamine biosynthesis in prostate cell lines, by blocking the activity of AMD1, reduced their dependence on folic acid in the media. This further strengthened the connection between polyamine metabolism and dependence on folic acid.
3. Prolonged growth of prostate cell lines with limited folic acid in the media resulted in alterations in nucleotide and SAM pools, but the same was not true in colorectal cancer cell lines.

While it is true, of course, that the 2009 study helped inform our plans and our ability to obtain funding for the current study, this in no way detracts from its novelty. The conclusions from the

2009 study are all about how dietary folate deficiency or supplementation may impact prostate cancer carcinogenesis. There are no data relevant to therapeutics, nor is there even any speculation as to the therapeutic implications. On the other hand, the current study describes an entirely new therapeutic strategy for prostate cancer. There are currently no metabolism based therapeutic strategies in use for treating prostate cancer outside of attacks on the androgen axis. Our strategy exploits a metabolic weakness of prostate cancer that is independent of the androgen axis. The following are some specific novel points of our current article:

1. We defined a novel synergistic relationship between activation of polyamine catabolism with a polyamine analogue and inhibition of the methionine salvage pathway with a transition state analogue. We demonstrate that this is effective in both androgen replete and zero androgen conditions.
2. We confirmed that SSAT is the relevant target of BENSpm by demonstrating its increased activity with BENSpm treatment and by showing that knockdown of SSAT rescued the antiproliferative effect of using BENSpm.
3. Additional mechanistic studies revealed that treatment led to reduction of polyamine and SAM pools and that this affect was more pronounced in the androgen dependent models.
4. We further defined the mechanism of action to lead to activation of PAOX and SMOX resulting in increased ROS and demonstrated that overexpression of TXNRD2, a ROS scavenger, reduced both the levels of ROS and the antiproliferative effect of treatment. We found that the ability of TXNRD2 to reduce the ROS effect was more pronounced in the C4-2 and CWR22RV1, which are grown in zero androgen conditions, than in the LNCaP line. We further explored why TXNRD2 overexpression failed to reduce the level of ROS that drug treatment induced in LNCaP cells and found that LNCaP and C4-2 cells have different antioxidant capacity and that TXNRD2 is able to rescue the effects of higher levels of ROS in C4-2.
5. We performed three months of toxicity studies in mice for the combination therapy of BENSpm and MTDIA and demonstrated no toxicity.
6. We further validated the efficacy of the combination therapy with an androgen independent xenograft grown in castrated male nude mice and found that the combination therapy is additive compared to either drug alone. We also tested two different dosing schedules and found one to be ineffective. Thus, these studies provide a novel combination therapy effective at blocking growth of androgen independent prostate cancer in castrate conditions, and optimize the dosing of BENSpm, which is an open question in the field.
7. These studies also explored *in vivo* mechanisms of action by demonstrating changes similar to those seen in the cell line models include depletion of polyamine pools and alterations of SAM/SAH pools and accumulation of 8-oxo-dG, an indicator of oxidative DNA damage suggesting a significant role played by ROS accumulation.
8. We demonstrated efficacy of the combination therapy in human *ex vivo* prostatectomy specimens by demonstrating an increase in a marker of apoptosis, increased abundance of the target enzyme SSAT which is predicted to be stabilized by BENSpm, and

decreases of intracellular polyamine pools in a dose dependent manner. Furthermore we demonstrate a direct and strong correlation between upregulation of the target enzyme SSAT and decrease in polyamine levels.

- For the third part of this statement, concerning lack of mechanism underlying increased ac-polyamine secretion in response to MTAP inhibition in the Rv1 cell line, we argue that this is a minor point in the overall context of the study. One of three cell lines shows a small increase in secretion of acetylated polyamines upon MTDIA treatment. We have speculated that accumulation of MTA, caused by MTAP inhibition, may block the activity of polyamine synthases and lead to a distortion of the spm:spd ratio. This may lead to a low level of activation of catabolism, initiated by acetylation of polyamines, as a means of rebalancing the spm:spd ratio. Some fraction of the acetylated polyamines may then be secreted before being acted on by PAOX. We have not pursued this line of questioning because we view it as a minor point in the overall context of the study occurring in only a single cell line. Furthermore, in all other experiments, we found minimal impact of single agent MTDIA. That we did not pursue this specific line of questioning does not take away from the other mechanisms investigated that we enumerate and describe above.